# Imbalances in Neurosymbolic Learning: Characterization and Mitigating Strategies

**Efthymia Tsamoura**†*
Huawei Labs
efthymia.tsamoura@huawei.com

**Kaifu Wang**†
University of Pennsylvania
kaifu@sas.upenn.edu

**Dan Roth**
University of Pennsylvania
danroth@seas.upenn.edu

## Abstract

We study one of the most popular problems in *neurosymbolic learning* (NSL), that of learning neural classifiers given only the result of applying a symbolic component $\sigma$ to the gold labels of the elements of a vector $\mathbf{x}$. The gold labels of the elements in $\mathbf{x}$ are unknown to the learner. We make multiple contributions, theoretical and practical, to address a problem that has not been studied so far in this context, that of characterizing and mitigating *learning imbalances*, i.e., major differences in the errors that occur when classifying instances of different classes (aka *class-specific risks*). Our theoretical reveals a unique phenomenon: that $\sigma$ can greatly impact learning imbalances. This result sharply contrasts with previous research on supervised and weakly supervised learning, which only studies learning imbalances under data imbalances. On the practical side, we introduce a technique for estimating the marginal of the hidden gold labels using weakly supervised data. Then, we introduce algorithms that mitigate imbalances at training and testing time by treating the marginal of the hidden labels as a constraint. We demonstrate the effectiveness of our techniques using strong baselines from NSL and long-tailed learning, suggesting performance improvements of up to 14%.

## 1 Introduction

The need to address the limitations of deep learning motivated researchers to explore *neurosymbolic learning* (NSL) [13], a family of techniques that integrate neural mechanisms for inference and learning with symbolic ones. This work considers one of the most popular NSL learning settings [12, 61, 20, 28, 40, 39, 37] in which a neural classifier $f$ is learned assuming access only to a vector of inputs $\mathbf{x} = (x_1, \ldots, x_M)$ to $f$ and to the result of applying $\sigma$ to the gold labels of the $x_i$s. The gold labels are hidden during learning. An example is illustrated below:

**Example 1.1** (Example adapted from [36]). *We aim to learn an MNIST classifier $f$, using only samples of the form $(x_1, x_2, s)$, where $x_1$ and $x_2$ are MNIST digits and $s$ is the maximum of their gold labels, i.e., $s = \sigma(y_1, y_2) = \max\{y_1, y_2\}$ with $y_i$ being the label of $x_i$. The gold labels are hidden during training. We will refer to the $y_i$'s and $s$ as* hidden *and* weak *labels, respectively.*

Our learning setting, which we will refer to as NESY, has been extensively adopted in NLP [58, 49, 47, 68, 16]. Recently, NESY has been successfully adopted to fine-tune language models [73, 29], align

---

*Work started before Efthymia Tsamoura joined Huawei Labs.
†These authors contributed equally to this work.

39th Conference on Neural Information Processing Systems (NeurIPS 2025).

video to text [21], perform visual question answering [20], and learn knowledge graph embeddings [34, 35]. The wide range of applications of NESY motivated its extensive study [67, 39, 40, 28].

We, for the first time, study an unexplored topic in the context of NESY: the characterization and mitigation of *learning imbalances*, i.e., the major differences in errors occurring when classifying instances of different classes (aka *class-specific risks*). Existing work on supervised [42, 6] and weakly supervised learning [65, 18] studies imbalances under the prism of *long-tailed* (aka *imbalanced*) data: data in which instances of different classes occur with very different frequencies, [17, 19, 4]. However, these results cannot fully characterize learning imbalances in NESY. This is because the *symbolic component $\sigma$ may cause learning imbalances even when the hidden or the weak labels are uniformly distributed*. Figure 1 demonstrates this phenomenon by showing the accuracy of the classification per class at different training epochs when an MNIST classifier is trained as in Example 1.1 and the hidden labels are uniform. Hence, to formalize the imbalances in NESY, we need to account for the symbolic component $\sigma$.

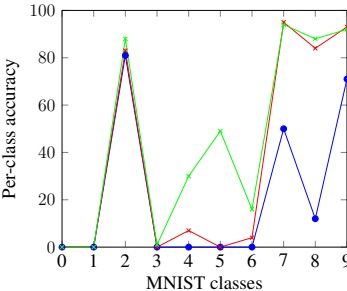

Figure 1: Class-specific accuracies of classifier $f$ (Example 1.1). Blue, red, and green curves show accuracy at 20, 40 and 100 epochs. Learning converges in 100 epochs.

On the practical side, mitigating learning imbalances (a problem typically referred to as *long-tailed learning*) has received considerable attention in supervised and weakly supervised learning with the proposed techniques operating at training [6, 60, 59, 9, 4] or at testing time [25, 46, 42]. However, these previous algorithms are not appropriate for NESY. First, they rely on (good) approximations of the marginal distribution of the hidden labels. Although approximating $\mathbf{r}$ may be easy in supervised learning [42] since the gold labels are available, in our setting the gold labels are hidden. Second, the state-of-the-art for training time mitigation [65, 6, 60, 59, 9, 4, 18] is designed for settings in which a single instance is presented each time to the learner and hence, they cannot take into account the correlations among the instances.

**Contributions.** We first provide class-specific error bounds in the context of NESY. Complementary to previous work in supervised learning [6] and weakly supervised one [10], our theory shows that $\sigma$ can significantly affect learning imbalances, see Theorem 3.1. Our analysis extends the theoretical analysis in [67] – by providing stricter error bounds and making fewer assumptions – and the theoretical analysis in [10].

We then propose a statistically consistent technique for estimating the marginals of the hidden labels given weak labels and two algorithms to mitigate imbalances during training and testing time. The first algorithm assigns pseudolabels to training data based on a novel linear programming formulation of NESY, see Section 4.2. The second algorithm uses the marginals of the hidden labels to constrain the model's predictions on test data using robust semi-constrained optimal transport [26], see Section 4.3. Our empirical analysis shows that our techniques can improve the accuracy over strong baselines in NSL [71, 67] and long-tailed learning [42, 18] by up to 14% and that the straightforward application of previous state-of-the-art to NESY is impossible [65] or problematic [18].

Proofs, additional backgrounds and details on our empirical analysis are in the appendix. The source code to run our empirical analysis are available at https://github.com/tsamoura/imbalances-nsl.

## 2 Preliminaries

Our notation is summarized in Table 5 and 6 and builds on [67, 28, 61].

**Data and models.** For an integer $n \geq 1$, let $[n] := \{1, \ldots, n\}$. Let also $\mathcal{X}$ be the instance space and $\mathcal{Y} = [c]$ be the output space. We use $x, y$ to denote elements in $\mathcal{X}$ and $\mathcal{Y}$. The distribution of two random variables $X, Y$ over $\mathcal{X} \times \mathcal{Y}$ is denoted by $\mathcal{D}$, and $\mathcal{D}_X, \mathcal{D}_Y$ denote the marginals of $X$ and $Y$. The vector $\mathbf{r} = (r_1, \ldots, r_c)$ denotes $\mathcal{D}_Y$, where $r_j := \mathbb{P}(Y = j)$ is the probability (or ratio) label $j \in \mathcal{Y}$ occurs in $\mathcal{D}$. We consider *scoring functions* $f$ that given instances from $\mathcal{X}$ output softmax probabilities (or *scores*). We use $f^j(x)$ to denote the score of $f$ for class $j \in \mathcal{Y}$. A scoring function $f$ induces a *classifier* $[f] : \mathcal{X} \to \mathcal{Y}$, whose *prediction* on $x$ is given by $\operatorname{argmax}_{j \in [c]} f^j(x)$. We denote

by $\mathcal{F}$ the set of scoring functions and by $[\mathcal{F}]$ the set of classifiers. The *zero-one risk* $R(f)$ of $f$ is the probability $f$ misclassifies an input instance. The *class-specific* of $f$ for class $j$ is the probability $f$ misclassifies an instance of that class, i.e., $R_j(f) := \mathbb{P}([f](x) \neq j | Y = j)$.

**Neurosymbolic learning.** We align with the notation from [12, 61, 28] and assume that each NESY training sample is of the form $(\mathbf{x}, s)$, where $\mathbf{x} = (x_1, \ldots, x_M)$ is a vector of instances in $\mathcal{X}^M$ and $s \in \mathcal{S}$ is the result of applying the symbolic component $\sigma$ over the hidden gold labels $\mathbf{y} = (y_1, \ldots, y_M)$ of the elements of $\mathbf{x}$. We assume that $\sigma$ is known to the learner, similarly to [20, 28, 40, 39]. As first notated in [61], using abduction [24], the symbolic component $\sigma$ can be seen as a function from $\mathcal{Y}^M$ to $\mathcal{S}$. We refer to $\mathcal{S} = \{a_1, \ldots, a_{c_S}\}$, where $|\mathcal{S}| = c_S \geq 1$, as the *space of weak labels* and to an element from $\mathcal{S}$ as a *weak label*. We denote the set of all label vectors that map to $s$ under $\sigma$ by $\sigma^{-1}(s)$. Each vector in $\sigma^{-1}(s)$ may be the gold vector of labels. Returning to Example 1.1, $\sigma^{-1}(s = 1) = \{(0, 1), (1, 0), (1, 1)\}$. We refer to each vector in $\sigma^{-1}(s)$ as a *pre-image*. The distribution of samples $(\mathbf{x}, s)$ is denoted by $\mathcal{D}_\mathsf{P}$. We denote a set of $m_\mathsf{P}$ NESY samples by $\mathcal{T}_\mathsf{P}$. We set $[f](\mathbf{x}) := ([f](x_1), \ldots, [f](x_M))$. The *zero-one partial loss* is defined as $L_\sigma(\mathbf{y}, s) := L(\sigma(\mathbf{y}), s) = \mathbb{1}\{\sigma(\mathbf{y}) \neq s\}$, for any $\mathbf{y} \in \mathcal{Y}^M$ and $s \in \mathcal{S}$. We aim to find the classifier $f$ with the minimum *zero-one partial risk* given by $R_\mathsf{P}(f; \sigma) := \mathbb{E}_{(X_1, \ldots, X_M, S) \sim \mathcal{D}_\mathsf{P}}[L_\sigma(([f](\mathbf{X})), S)]$.

Relevant NSL work [37, 20, 40] may denote training samples differently. However, this notation is equivalent with ours, see Appendix A. Furthermore, our definition of NESY aligns with that of *multi-instance partial label learning* (MI-PLL) without assuming that the $\mathcal{X}$ instances in $\mathcal{D}_\mathsf{P}$ are i.i.d. As discussed in [67], *partial label learning* (PLL) [10, 5, 70], where each training instance is associated with a set of mutually exclusive candidate labels, is a special case of NESY, see Appendix E.

**Vectors and matrices.** A vector $\mathbf{v}$ is *diagonal* if all of its elements are equal. We denote by $\mathbf{e}_i$ the one-hot vector, where the $i$-th element equals 1. We denote the all-one and the all-zero vectors by $\mathbf{1}_n$ and $\mathbf{0}_n$, and the identity matrix of size $n \times n$ by $\mathbf{I}_n$. Let $\mathbf{A} \in \mathbb{R}^{n \times m}$ be a matrix. We use $A_{i,j}$ to denote the value of the $(i, j)$ cell of $\mathbf{A}$ and $v_i$ to denote the $i$-th element of $\mathbf{v}$. The *vectorization* of $\mathbf{A}$ is given by $\mathrm{vec}(\mathbf{A}) := [a_{1,1}, \ldots, a_{n,1}, \ldots, a_{1,m}, \ldots, a_{n,m}]^\mathsf{T}$ and its *Moore–Penrose inverse* by $\mathbf{A}^\dagger$. If $\mathbf{A}$ is square, then the diagonal matrix that shares the same diagonal with $\mathbf{A}$ is denoted by $D(\mathbf{A})$. For matrices $\mathbf{A}$ and $\mathbf{B}$, $\mathbf{A} \otimes \mathbf{B}$ and $\langle \mathbf{A}, \mathbf{B} \rangle$ denote their *Kronecker* and *Frobenius inner products*.

## 3 Theory: Characterizing Learning Imbalances In NESY

We provide error bounds that measure the difficulty of learning instances of each class in $\mathcal{Y}$ using NESY data. The bounds indicate that, unlike supervised learning, *learning imbalances in* NESY *arise not only from imbalances in the hidden or weak label distributions, but also from the symbolic component $\sigma$*. Our analysis is based on the assumption that the $\mathcal{X}$ instances in $\mathcal{D}_\mathsf{P}$ are i.i.d. To simplify the presentation, our analysis focuses on $M = 2$. However, it can be generalized to $M > 2$.

Our theory is based on a novel nonlinear program formulation that allows us to compute an upper bound of each $R_j(f)$. The first key idea (K1) to that formulation is a rewriting of $R_\mathsf{P}(f; \sigma)$ and $R_j(f)$. To start with, given the function $\sigma$, the zero-one partial risk can be expressed as

probability of the label pair $(i,j)$

the weak label is misclassified

$$R_\mathsf{P}(f; \sigma) = \sum_{(i,j) \in \mathcal{Y}^2} r_i r_j \left( \sum_{(i',j') \in \mathcal{Y}^2} \mathbb{1}\{\sigma(i,j) \neq \sigma(i',j')\} \, \mathbf{H}_{ii'}(f) \mathbf{H}_{jj'}(f) \right) \quad (1)$$

conditional probability that the labels $i$ and $j$ are (mis)classified as $i'$ and $j'$

where $\mathbf{H}(f)$ is an $c \times c$ matrix defined as $\mathbf{H}(f) := [\mathbb{P}([f](x) = j | Y = i)]_{i \in [c], j \in [c]}$. To derive (1), we enumerate all the 4-ary vectors $(i, j, i', j') \in \mathcal{Y}^4$, where $i, j$ are the gold hidden labels and $i', j'$ are the predicted labels, so that the predicted labels lead to a wrong weak label, i.e., $\sigma(i, j) \neq \sigma(i', j')$. The risk $R_\mathsf{P}(f; \sigma)$ is the sum of the probabilities of those wrong predictions, with $H_{ii'}(f)H_{jj'}(f)$ encoding the probability of occurrence of the vectors $(i, j, i', j')$. Now, let $\mathbf{h}(f)$ be the vectorization of $\mathbf{H}(f)$. The partial risk $R_\mathsf{P}(f; \sigma)$ in (1) is a quadratic form of $\mathbf{h}(f)$. Therefore, there is a unique symmetric matrix $\mathbf{\Sigma}_{\sigma, \mathbf{r}}$ in $\mathbb{R}^{c^2 \times c^2}$ that depends only on $\sigma$ and $\mathbf{r}$ such that (1) can be rewritten as $R_\mathsf{P}(f; \sigma) = \mathbf{h}(f)^\mathsf{T} \mathbf{\Sigma}_{\sigma, \mathbf{r}} \mathbf{h}(f)$. Furthermore, for each $j \in \mathcal{Y}$, let $\mathbf{W}_j$ be the matrix defined by $(\mathbf{1}_c - \mathbf{e}_j)\mathbf{e}_j^\mathsf{T}$, and $\mathbf{w}_j$ be its vectorization. We can rewrite the class-specific risk as $R_j(f) = \mathbf{w}_j^\mathsf{T} \mathbf{h}(f)$.

The second key idea (K2) to form a nonlinear program that computes class-specific risk bounds is to upper bound the class-specific risk $R_j(f)$ of a model $f$ with the model's partial risk $R_\mathsf{P}(f; \sigma)$.

The latter can be minimized with NeSy training data $\mathcal{T}_{\mathsf{P}}$. Putting (K1) and (K2) together, the worst class-specific risk of $f$ for class $j \in \mathcal{Y}$ is given by the optimal solution to the program below:

$$
\begin{aligned}
\max_{\mathbf{h}} \quad & \mathbf{w}_j^{\mathsf{T}} \mathbf{h}(f) \\
\text{s.t.} \quad & \mathbf{h}(f)^{\mathsf{T}} \mathbf{\Sigma}_{\sigma,\mathbf{r}} \mathbf{h}(f) = R_{\mathsf{P}}(f;\sigma) && \text{(partial risk)} \\
& \mathbf{h}(f) \geq 0 && \text{(positivity)} \\
& (\mathbf{I}_c \otimes \mathbf{1}_c^{\mathsf{T}}) \mathbf{h}(f) = \mathbf{1}_c && \text{(normalization)}
\end{aligned}
\tag{2}
$$

Let us analyze (2). The optimization objective states that our aim is to find the worst possible class-specific risk, expressed as $R_j(f) = \mathbf{w}_j^{\mathsf{T}} \mathbf{h}(f)$. The first constraint specifies the partial risk of the model. The second one requires the (mis)classification probabilities to be nonnegative. The last constraint, where $(\mathbf{I}_c \otimes \mathbf{1}_c^{\mathsf{T}}) \mathbf{h}(f)$ represents the row sums of matrix $\mathbf{H}(f)$, enforces the classification probabilities to sum to one. Let $\Phi_{\sigma,j}(R_{\mathsf{P}}(f;\sigma))$ denote the optimal solution of program (2). We have:

**Proposition 3.1** (Class-specific risk bound). *For any $j \in \mathcal{Y}$, we have that $R_j(f) \leq \Phi_{\sigma,j}(R_{\mathsf{P}}(f;\sigma))$.*

**Characterizing learning imbalances.** Proposition 3.1 suggests that the worst risk associated with each class in $\mathcal{Y}$ is characterized by two factors: (i) the model's partial risk $R_{\mathsf{P}}(f;\sigma)$, which is independent of the specific class and (ii) $\sigma$, since $\sigma$ affects the mapping $\Phi_{\sigma,j}$ from the model's partial risk to the class-specific risk. Therefore, the learning imbalance can be assessed by comparing the growth rates of $\Phi_{\sigma,j}$. We use this approach to analyze Example 1.1.

**Example 3.2** (Cont' Example 1.1). *Let $\mathcal{D}$ and $\mathcal{D}_{\mathsf{P}}$ be defined as in Section 2. Consider the two cases:*

CASE 1 *The marginal of the hidden label $Y$ is uniform. The left-hand side of Figure 2 shows the risk bounds for different classes obtained by solving the program (2). The bounds are presented as functions of the different values of $R_{\mathsf{P}}(f;\sigma)$. In this plot, the curve for class "zero" (resp. "nine") has the steepest (resp. smoothest) slope, suggesting that $f$ will tend to make more (resp. fewer) mistakes when classifying instances of that class. In other words, class "zero" is the hardest to learn, as also shown to be the case in reality, see Figure 1.*

CASE 2 *The marginal of the weak label $S$ is uniform. Similarly, the right-hand side of Figure 2 plots the corresponding risk bounds, suggesting that the class "zero" is now the easiest to learn.*

**Computable bounds.** Via Proposition 3.1, we can derive a bound for $R_j(f)$ that can be computed using a NeSy dataset. This can be done by using standard learning theory tools (e.g., the VC-dimension or the Rademacher complexity) to show that, given a fixed confidence level $\delta \in (0,1)$, the partial risk $R_{\mathsf{P}}(f;\sigma)$ will

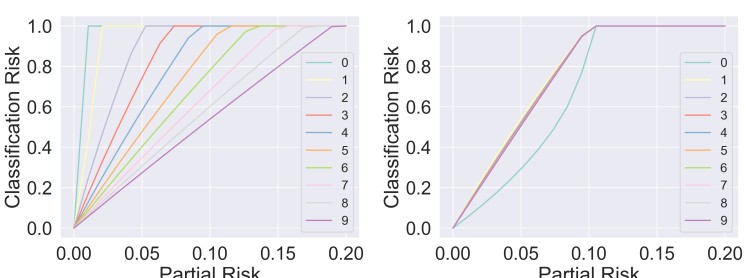

Figure 2: Class-specific upper bounds obtained via (2). (left) $\mathcal{D}_Y$ is uniform. (right) $\mathcal{D}_{\mathsf{P}_S}$ is uniform.

not exceed a *generalization bound* $\widetilde{R}_{\mathsf{P}}(f;\sigma, \mathcal{T}_{\mathsf{P}}, \delta)$ with probability $1 - \delta$:

**Proposition 3.3.** *Let $d_{[\mathcal{F}]}$ be the Natarajan dimension of $[\mathcal{F}]$. Given a confidence level $\delta \in (0,1)$, we have that $R_j(f) \leq \Phi_{\sigma,j}(\widetilde{R}_{\mathsf{P}}(f;\sigma, \mathcal{T}_{\mathsf{P}}, \delta))$ with probability $1 - \delta$ for any $j \in [c]$, where*

$$
\widetilde{R}_{\mathsf{P}}(f;\sigma, \mathcal{T}_{\mathsf{P}}, \delta) = \widehat{R}_{\mathsf{P}}(f;\sigma, \mathcal{T}_{\mathsf{P}}) + \sqrt{\frac{2\log(em_{\mathsf{P}}/2d_{[\mathcal{F}]}\log(6Mc^2 d_{[\mathcal{F}]}/e))}{m_{\mathsf{P}}/2d_{[\mathcal{F}]}\log(6Mc^2 d_{[\mathcal{F}]}/e)}} + \sqrt{\frac{\log(1/\delta)}{2m_{\mathsf{P}}}}
\tag{3}
$$

The first term on the right-hand side of (3) denotes the empirical partial risk of classifier $f$, the second term upper bounds the Natarajan dimension of $f$ [55], and the third term quantifies the confidence level or the probability that the generalization bound holds, which is typical in learning theory. The Proposition 3.3 shows the speed of decrease of the risk of $f$ for class $j \in \mathcal{Y}$ when using NeSy samples for training. Further on our bounds and Example 3.2 are in Appendix B.2.

**Comparison to previous work.** Our result extends [67] (see Section 2 for a discussion about MI-PLL and NeSy) in three ways: (i) we bound the risks $R_j(f)$ instead of bounding the total risk $R(f)$; (ii)

our bounds do not rely on $M$-unambiguity, in contrast to those in [67]; and (iii) the program (2) leads to tighter bounds for $R(f)$. Before proving (iii), let us first recapitulate $M$-unambiguity [67], where a function $\sigma$ is $M$-*unambiguous* if for any two diagonal label vectors $\mathbf{y}$ and $\mathbf{y}' \in \mathcal{Y}^M$ such that $\mathbf{y} \neq \mathbf{y}'$, we have that $\sigma(\mathbf{y}') \neq \sigma(\mathbf{y})$. Now, let us move to point (iii). By relaxing the constraints in (2), we can recover Lemma 1 from [67] (which is the key to proving Theorem 1 from [67]). In particular, if we: (i) drop the positivity and normalization constraints from (2) and (ii) replace the partial risk constraint by the more relaxed inequality $\mathbf{h}(f)^\mathsf{T} D(\boldsymbol{\Sigma}_{\sigma,\mathbf{r}}) \mathbf{h}(f) \leq R_\mathsf{P}(f;\sigma)$, we obtain the following:

**Proposition 3.4.** *If $\sigma$ is $M$-unambiguous, we have*

$$R(f) \leq \sqrt{\mathbf{w}^\mathsf{T}(D(\boldsymbol{\Sigma}_{\sigma,\mathbf{r}}))^\dagger \mathbf{w} R_\mathsf{P}(f;\sigma)} = \sqrt{c(c-1)R_\mathsf{P}(f;\sigma)} \qquad (4)$$

*which coincides with Lemma 1 from [67] for $M = 2$, where $\mathbf{w} := \sum_{j=1}^c r_j \mathbf{w}_j$.*

## 4 Algorithms: Mitigating Imbalances In NeSy

Section 3 sends a clear message: NeSy is prone to learning imbalances that may be exacerbated due to $\sigma$. The results of our theoretical analysis motivate us to develop a portfolio of techniques to address learning imbalances. Our first contribution, see Section 4.1, is a statistically consistent technique for estimating $\mathbf{r}$, assuming access to weak labels only. We then proceed with training and testing time mitigation. Our mitigation algorithms enforce the class priors to a classifier's predictions, a common idea in long-tailed learning. The intuition is that the classifier will tend to predict the labels that appear more often in the training data. Hence, enforcing the priors gives more importance to the minority classes at training time and encourages the model to predict minority classes at testing time. Our marginal estimation algorithm requires the assumption that the $\mathcal{X}$ instances in $\mathcal{D}_\mathsf{P}$ are i.i.d.; the other algorithms work even when this assumption fails. Table 6 summarizes the notation in Section 4.

### 4.1 Estimating The Marginal Of The Hidden Labels

We begin with our technique for estimating $\mathbf{r}$ using only NeSy data $\mathcal{T}_\mathsf{P}$. We denote the probability of occurrence (or ratio) of the $j$-th weak label $a_j \in \mathcal{S}$ by $p_j := \mathbb{P}(S = a_j)$ and set $\mathbf{p} = (p_1, \ldots, p_{c_S})$. To estimate $\mathbf{r}$, we rely on the observation that in NeSy, $p_j$ equals the probability of the label vectors in $\sigma^{-1}(a_j)$, namely $p_j = \sum_{(y_1,\ldots,y_M) \in \sigma^{-1}(a_j)} \prod_{i=1}^M r_{y_i}$, which is a polynomial of $\mathbf{r}$.

**Example 4.1.** *Consider* Case (2) *from Example 3.2. Assume that the marginals of the weak labels are uniform. Then, we can obtain $\mathbf{r}$ by solving the following* system of polynomial equations: $[r_0^2, r_1^2 + 2r_0r_1, \ldots, r_9^2 + 2\sum_{i=0}^8 r_i r_9]^\mathsf{T} = [1/10, 1/10, \ldots, 1/10]^\mathsf{T}$. *The first equation denotes the probability a weak label is zero, which is $1/10$ (uniformity). Due to $\sigma$, this can happen only when $y_1 = y_2 = 0$. Under the independence assumption, the above implies that $r_0^2 = 1/10$. Analogously, the second and the last polynomials denote the probability a weak label is one and nine.*

Let $P_\sigma$ be the system of polynomials $[p_j]_{j \in [c_S]}^\mathsf{T} = [\sum_{(y_1,\ldots,y_M) \in \sigma^{-1}(a_j)}]_{j \in [c_S]}^\mathsf{T}$. Let $\Psi_\sigma$ be the function mapping each $r_j \in \mathcal{Y}$ to its solution in $P_\sigma$, assuming $\mathbf{p}$ is known. In practice, $\mathbf{p}$ is unknown, but can be estimated from a NeSy dataset $\mathcal{T}_\mathsf{P}$, namely $\bar{p}_j := \sum_{k=1}^{|\mathcal{T}_\mathsf{P}|} \mathbb{1}\{s_k = a_j\}/|\mathcal{T}_\mathsf{P}|$. As the $\bar{p}_j$'s can be noisy, the system of polynomials could become inconsistent. Therefore, instead of solving the polynomial equation as in Example 4.1, we find an estimate $\widehat{\mathbf{r}}$, so that its induced prediction for the weak label ratio $\widehat{\mathbf{p}} := \Psi_\sigma(\widehat{\mathbf{r}})$ best fits to the empirical probabilities $\bar{p}_j$'s by means of cross-entropy. Since this requires optimizing over the probability simplex $\Delta_c$, we reparametrize the estimated ratios $\widehat{\mathbf{r}}$ by $\mathrm{softmax}(\mathbf{u})$, leading to the Algorithm 1. We prove its consistency in Appendix C.

### 4.2 Training Time Imbalance Mitigation Via Linear Programming

We now turn to training time mitigation. We aim to find pseudolabels $\mathbf{Q}$ that are close to the classifier's scores and adhere to $\widehat{\mathbf{r}}$ and use $\mathbf{Q}$ to train the classifier using the cross-entropy loss. There are two design choices: (i) whether to find pseudolabels at the individual instance level or at the batch level; (ii) whether to be strict in enforcing the marginal $\widehat{\mathbf{r}}$. In addition, we face two challenges: (iii) we are provided $M$-ary tuples of instances of the form $(x_1, \ldots, x_M)$; (iv) $\mathbf{Q}$ must additionally abide by the constraints coming from $\sigma$ and the weak labels, e.g., when $s = 1$ in Example 1.1, then the only valid label assignments for $(x_1, x_2)$ are (1,1), (0,1) and (1,0). Regarding (i), finding pseudolabels at

| **Algorithm 1** LABEL RATIO SOLVER | **Algorithm 2** CAROT |
|---|---|
| **Input:** weak labels $\{s_k\}_{k=1}^{m_P}$, function $\sigma$, step size $t$, iterations $N_{\text{iter}}$ | **Input:** model's raw scores $\mathbf{P} \in \mathbb{R}^{c \times n}$, ratio estimates $\widehat{\mathbf{r}} \in \mathbb{R}^c$, entropic reg. parameter $\eta > 0$, margin reg. parameter $\tau > 0$, iterations $N_{\text{iter}}$ |
| **Initialize:** logit $\mathbf{u} \leftarrow \mathbf{1}_c$; $\bar{p}_j$, for $j \in [c_S]$ | **Initialize:** $\mathbf{u} \leftarrow \mathbf{0}_n$; $\mathbf{v} \leftarrow \mathbf{0}_c$ |
| **for** $N = 1, \ldots, N_{\text{iter}}$ **do** | **for** $N = 1, \ldots, N_{\text{iter}}$ **do** |
| $\quad \widehat{\mathbf{r}} \leftarrow \text{softmax}(\mathbf{u})$ | $\quad \mathbf{a} \leftarrow B(\mathbf{u}, \mathbf{v})\mathbf{1}_c; \quad \mathbf{b} \leftarrow B(\mathbf{u}, \mathbf{v})^{\mathsf{T}}\mathbf{1}_n$ |
| $\quad$ **for each** $j \in [c_S]$ **do** | $\quad$ **if** $k$ is even **then** |
| $\quad\quad \widehat{p}_j \leftarrow \sum\limits_{(y_1, \ldots, y_M) \in \sigma^{-1}(a_j)} \prod_{i=1}^{M} \widehat{r}_{y_i}$ | $\quad\quad$ **update v** //see Section 4.3 |
| $\quad \ell \leftarrow \sum_{j=1}^{c_S} \bar{p}_j \log \widehat{p}_j$ | $\quad$ **else** |
| $\quad$ Backpropagate $\ell$ to update $\mathbf{u}$ | $\quad\quad$ **update u** //see Section 4.3 |
| **return** $\text{softmax}(\mathbf{u})$ | **return** $B(\mathbf{u}, \mathbf{v})$ |

the individual instance level does not guarantee that the modified scores match $\widehat{\mathbf{r}}$ [46]. Regarding (ii), strictly enforcing $\widehat{\mathbf{r}}$ could be problematic as $\widehat{\mathbf{r}}$ can be noisy.

To accommodate the above requirements while avoiding the crux of solving nonlinear programs, we rely on a novel *linear programming* (LP) formulation of NESY that finds pseudolabels for a batch of $n$ scores. We use $(x_{\ell,1}, \ldots, x_{\ell,M}, s_\ell)$ to denote the $\ell$-th NESY training sample in a batch of size $n$. We also use $\mathbf{P}_i \in [0, 1]^{n \times c}$ and $\mathbf{Q}_i \in [0, 1]^{n \times c}$, for $i \in [M]$, to denote the classifier's scores and the pseudolabels assigned to the $i$-th input instances of the batch. In particular, $P_i[\ell, j] = f^j(x_{\ell,i})$, while $Q_i[\ell, j]$ is the corresponding pseudolabel. Before continuing, it is crucial to explain how to associate each training sample $s_\ell$ with a Boolean formula in *disjunctive normal form* (DNF). Associating weak labels with DNF formulas is standard in the neurosymbolic literature [71, 61, 20, 67]. For $\ell \in [n]$, $i \in [M]$, and $j \in [c]$, let $q_{\ell,i,j}$ be a Boolean variable that is true if $x_{\ell,i}$ is assigned label $j \in \mathcal{Y}$ and false otherwise. Let $R_\ell$ be the size of $\sigma^{-1}(s_\ell)$. Based on the above, we can associate each label vector $\mathbf{y}$ in $\sigma^{-1}(s_\ell)$ with a conjunction $\phi_{\ell,t}$ of Boolean variables from $\{q_{\ell,i,j}\}_{i \in [M], j \in [c]}$, such that $q_{\ell,i,j}$ occurs in $\phi_{\ell,t}$ only if the $i$-th label in $\mathbf{y}$ is $j \in \mathcal{Y}$. We assume a canonical ordering over the variables occurring in each $\varphi_{\ell,t}$, for $t \in [R_\ell]$, and use $\varphi_{\ell,t,k}$ to refer to the $k$-th variable. We use $|\varphi_{\ell,t}|$ to denote the number of variables in $\varphi_{\ell,t}$.

Based on the above, finding a pseudolabel assignment for $(x_{\ell,1}, \ldots, x_{\ell,M})$ that adheres to $\sigma$ and $s_\ell$ reduces to finding an assignment to the variables in $\{q_{\ell,i,j}\}_{i \in [M], j \in [c]}$ that makes $\Phi_\ell$ hold. Previous work [50, 57] has shown that we can cast satisfiability problems to linear programming problems. Therefore, instead of finding a Boolean assignment to each $q_{\ell,i,j}$, we can find an assignment in $[0, 1]$ for the real counterpart of $q_{\ell,i,j}$ denoted by $[q_{\ell,i,j}]$. Via associating the $[q_{\ell,i,j}]$'s to the entries in the $\mathbf{Q}_i$'s, i.e., $Q_i[\ell, j] = [q_{\ell,i,j}]$, we can solve the following linear program to perform pseudolabeling:

$$\textbf{objective} \quad \min_{(\mathbf{Q}_1, \ldots, \mathbf{Q}_M)} \sum_{i=1}^{M} \langle -\log(\mathbf{P}_i), \mathbf{Q}_i \rangle,$$

$$\textbf{s.t.} \quad
\begin{aligned}
\sum_{t=1}^{R_\ell} [\alpha_{\ell,t}] &\geq 1, & \ell &\in [n] \\
-|\varphi_{\ell,t}|[\alpha_{\ell,t}] + \sum_{k=1}^{|\varphi_{\ell,t}|} [\varphi_{\ell,t,k}] &\geq 0, & \ell &\in [n], t \in [R_\ell] \\
-\sum_{k=1}^{|\varphi_{\ell,t}|} [\varphi_{\ell,t,k}] + [\alpha_{\ell,t}] &\geq (1 - |\varphi_{\ell,t}|), & \ell &\in [n], t \in [R_\ell] \\
\sum_{j=1}^{c} [q_{\ell,i,j}] &= 1, & \ell &\in [n], i \in [M] \\
[q_{\ell,i,j}] &\in [0, 1], & \ell &\in [n], i \in [M], j \in [c] \\
|\mathbf{Q}_i \cdot \mathbf{1}_n - n\widehat{\mathbf{r}}| &\leq \epsilon, & i &\in [M]
\end{aligned}
\tag{5}$$

The objective in (5) aligns with our aim to find pseudolabels close to the classifier's scores. The independence among the classifier's scores for different $\mathbf{x}_{\ell,i}$'s justifies the sum over different $i$'s in the minimization objective. The first three constraints force the pseudolabels for the $\ell$-th training sample to adhere to $\sigma$ and $s_\ell$, where the $\alpha_{\ell,t}$'s are Boolean variables introduced due to converting the $\Phi_\ell$'s into *conjunctive normal form* using the Tseytin transformation [62]. The fourth and the fifth constraint wants the pseudolabels for each instance $x_{\ell,i}$ to sum up to one and lie in $[0, 1]$. Finally, the last constraint wants for each $i \in [M]$, the probability of predicting the $j$-th pseudolabel for an element in $\{x_{\ell,i}\}_{\ell \in [n]}$ to match the ratio estimates at hand $\widehat{r}_j$ up to some $\epsilon \geq 0$: the smaller $\epsilon$ gets, the stricter the adherence to $\widehat{\mathbf{r}}$ becomes. The detailed derivation of (5) and an example are in Appendix D.

In summary, in training time mitigation, for each epoch, we split the training samples into batches. Then, for each batch $\{(x_{\ell,1}, \ldots, x_{\ell,M}, s_\ell)\}_{\ell \in [n]}$, we form $\mathbf{P}_1, \ldots, \mathbf{P}_M$ by applying $f$ to the $x_{\ell,i}$'s and solve (5) to get the pseudolabels $\mathbf{Q}_1, \ldots, \mathbf{Q}_M$. Finally, we minimize the cross-entropy loss between $\mathbf{Q}_1, \ldots, \mathbf{Q}_M$ and $\mathbf{P}_1, \ldots, \mathbf{P}_M$. We name this training technique LP. Our formulation in (5) is oblivious to $\widehat{\mathbf{r}}$, which can be estimated using Algorithm 1 or any other technique, e.g., [65].

### 4.3 CAROT: Testing Time Imbalance Mitigation

We conclude with CAROT, our algorithm to mitigate learning imbalances at testing time by modifying the model's scores to adhere to the estimated ratios $\widehat{\mathbf{r}}$. Incorporating $\widehat{\mathbf{r}}$ into the model's scores involves the design choices (i) and (ii) presented at the beginning of Section 4.2– challenges (iii) and (iv) are specific to training. Regarding (i), most existing testing time mitigation algorithms (e.g., [42]) modify a model's scores at the level of individual instances. Regarding (ii), as explained in Section 4.2, strictly enforcing $\widehat{\mathbf{r}}$ may also be problematic, since $\widehat{\mathbf{r}}$ may be different from the label marginals underlying the test data. Similarly to Section 4.2, we propose to adjust the model's scores for a whole batch of $n > 1$ test samples (represented by a matrix $\mathbf{P} \in \mathbb{R}^{n \times c}$) so that the adjusted scores $\mathbf{P}'$ roughly adhere to $\widehat{\mathbf{r}}$. Precisely, we propose to find the $\mathbf{P}'$ that optimizes the following objective:

$$\min_{\mathbf{P}' \in \mathbb{R}_+^{n \times c}, \mathbf{P}'\mathbf{1}_c = \mathbf{1}_n} \langle -\log(\mathbf{P}), \mathbf{P}' \rangle + \tau \, \mathrm{KL}(\mathbf{P}'^\mathsf{T} \mathbf{1}_n \| n\widehat{\mathbf{r}}) - \eta H(\mathbf{P}') \tag{6}$$

The first term in (6) encourages $\mathbf{P}'$ to be close to the original scores. The second term encourages the column sums of $\mathbf{P}'$ to match $\widehat{\mathbf{r}}$, with $\tau > 0$ controlling adherence, where KL is the Kullback-Leibler divergence. This formulation leads to a *robust semi-constrained optimal transport* (RSOT) problem [26]. The regularizer $\eta H(\mathbf{P}')$, where $H$ denotes entropy, allows to approximate the optimal solution using the robust semi-Sinkhorn algorithm [26], leading to CAROT (*Confidence-Adjustment via Robust semi-constrained Optimal Transport*), see Algorithm 2.

In Algorithm 2, $B(\mathbf{u}, \mathbf{v})$ denotes an $n \times c$ matrix whose $(i, j)$ cell is computed as a function of $\mathbf{u}$ and $\mathbf{v}$ by $\exp(u_i + v_j + \log(P_{ij})/\eta)$. In each iteration, the algorithm alternates between updating the $c$-dimensional vector $\mathbf{v}$ and the $n$-dimensional vector $\mathbf{u}$. The former update, which is computed as $\mathbf{v} \leftarrow \frac{\eta\tau}{\eta+\tau} \left( \frac{\mathbf{v}}{\eta} + \log(n\widehat{\mathbf{r}}) - \log(\mathbf{b}) \right)$, forces $B(\mathbf{u}, \mathbf{v})$ to adhere to $\widehat{\mathbf{r}}$; the latter, which is computed as $\mathbf{u} \leftarrow \eta \left( \frac{\mathbf{u}}{\eta} + \log(\mathbf{1}_n) - \log(\mathbf{a}) \right)$, forces the elements in each row of $B(\mathbf{u}, \mathbf{v})$ to add up to one.

**Choice of $\eta$ and $\tau$.** In practice, we use a small NESY validation set to choose $\eta$ and $\tau$. By doing so, the validation set can be obtained by splitting the training set of NESY data $\mathcal{T}_{\mathsf{P}}$.

**Guarantees.** The matrix $B(\mathbf{u}, \mathbf{v})$ converges to the optimal solution to (6) as $N_{\mathrm{iter}} \to \infty$, see [26].

## 5 Experiments

We consider the state-of-the-art loss *semantic loss* (SL) [71, 67, 20] and use the engine Scallop [20] that performs NESY training using that loss. We do not consider [12, 28, 61, 37, 38, 72] for reasons related to scalability (see [67, 20]), while the work in [40, 39] is orthogonal to ours. Since there are no prior NESY techniques for mitigating imbalances at testing time, we consider Logit Adjustment (LA) [42] as a competitor to CAROT. The notation +A, for $A \in \{\mathrm{LA}, \mathrm{CAROT}\}$, means that the scores of a baseline model are modified at testing time via A. We do not assume access to a validation set of gold labelled data, applying LA and CAROT using the estimate $\widehat{\mathbf{r}}$ obtained via Algorithm 1. We also run experiments with RECORDS [18], a technique that mitigates imbalances at training time for PLL [10] (no previous NESY training time baseline exists). We use SL+RECORDS when a classifier has been trained using RECORDS in conjunction with SL. RECORDS acts as a competitor to LP. Finally, we run experiments using LP, see Section 4.2. We use LP(ALG1) and LP(EMP), when LP is applied using the ratios obtained using Algorithm 1 and the approximation from [65].

**Benchmarks.** We carry experiments using NESY benchmarks previously used in the NSL literature [36, 38, 20, 28], namely MAX-$M$, SUM-$M$ [36, 20] and HWF-$M$ [28, 30], as well as a newly introduced, called Smallest Parent. Training samples in MAX-$M$ are as described in Example 1.1. We vary $M$ to $\{3, 4, 5\}$ and use the MNIST benchmark to obtain training and testing instances. In Smallest Parent, training samples are of the form $(x_1, x_2, p)$, where $x_1$ and $x_2$ are CIFAR-10 images and $p$ is the most immediate common ancestor of $y_1$ and $y_2$, assuming the classes form a hierarchy. To simulate long-tail phenomena (denoted as **LT**), we vary the imbalance ratio $\rho$ of the distributions

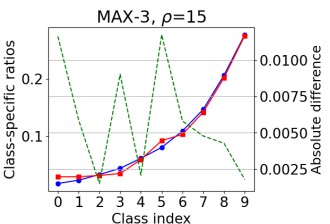 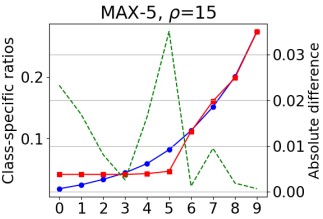 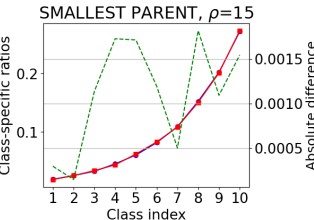

Figure 4: Accuracy of the marginal estimates computed by Algorithm 1. Blue denotes the gold ratios, red the estimated ones, and green the absolute difference between the gold and estimated ratios.

of the input instances as in [6, 65]: $\rho = 0$ means that the hidden label distribution is unmodified and balanced. Our scenarios are quite challenging. First, the pre-image of $\sigma$ may be particularly large, making the supervision rather weak, e.g., in the MAX-5 scenario, there are $5 \times 9^4$ candidate label vectors when the weak label is 9. Second, the functions may exacerbate the imbalances in the hidden labels, with the probability of certain weak labels getting very close to zero. For example, in MAX-5, the probability of $s = 0$ is $10^{-5}$ when $\rho = 0$. This probability becomes even smaller when $\rho = 50$. The results of our analysis are summarized in Table 1 and Figure 4. The accuracies in all the tables (obtained over three different for low-variance scenarios and ten runs over high-variance scenarios) are balanced, i.e., they are the weighted sums of the class-specific accuracies, where each weight is the ratio of the corresponding class in the test data. Due to lack of space, we discuss the results on SUM-$M$, HWF-$M$, and further details in Appendix F.

Table 1: (Top) Results for MAX-$M$ & $m_P = 3$K. (Bottom) Results for Smallest Parent & $m_P = 10$K.

| Algorithms | Original $\rho = 0$ | | | LT $\rho = 5$ | | | LT $\rho = 50$ | | |
| | $M = 3$ | $M = 4$ | $M = 5$ | $M = 3$ | $M = 4$ | $M = 5$ | $M = 3$ | $M = 4$ | $M = 5$ |
| --- | --- | --- | --- | --- | --- | --- | --- | --- | --- |
| SL | $84.15 \pm 11.92$ | $73.82 \pm 2.36$ | $59.88 \pm 5.58$ | $55.48 \pm 23.23$ | $66.24 \pm 1.22$ | $55.13 \pm 4.20$ | $66.74 \pm 5.42$ | $70.33 \pm 6.58$ | $55.74 \pm 2.58$ |
| + LA | $84.17 \pm 11.95$ | $73.82 \pm 2.36$ | $59.88 \pm 5.58$ | $55.48 \pm 23.23$ | $65.63 \pm 1.75$ | $55.13 \pm 4.20$ | $66.57 \pm 5.09$ | $61.10 \pm 3.95$ | $52.47 \pm 8.06$ |
| + CAROT | $84.57 \pm 11.50$ | $73.08 \pm 3.10$ | $60.26 \pm 5.20$ | $56.52 \pm 21.70$ | $66.70 \pm 0.76$ | $55.91 \pm 3.42$ | $68.16 \pm 4.00$ | $68.25 \pm 6.14$ | $57.29 \pm 14.17$ |
| RECORDS | $85.56 \pm 7.25$ | $75.11 \pm 0.77$ | $59.43 \pm 6.61$ | $77.98 \pm 3.13$ | $65.85 \pm 0.62$ | $55.07 \pm 4.24$ | $70.20 \pm 7.65$ | $72.05 \pm 8.34$ | $59.93 \pm 4.86$ |
| + LA | $87.63 \pm 5.11$ | $75.11 \pm 0.77$ | $59.28 \pm 6.76$ | $77.98 \pm 3.13$ | $65.43 \pm 0.87$ | $54.40 \pm 4.44$ | $70.09 \pm 7.26$ | $69.78 \pm 11.01$ | $59.93 \pm 4.86$ |
| + CAROT | $90.97 \pm 2.03$ | $75.94 \pm 0.91$ | $60.45 \pm 7.78$ | $78.31 \pm 4.00$ | $67.57 \pm 1.74$ | $55.46 \pm 3.94$ | $71.46 \pm 6.4$ | $71.25 \pm 8.70$ | $63.64 \pm 5.92$ |
| LP(EMP) | $94.97 \pm 1.32$ | $77.86 \pm 4.22$ | $55.27 \pm 11.27$ | $80.15 \pm 1.69$ | $70.73 \pm 1.85$ | $56.28 \pm 2.03$ | $77.16 \pm 3.46$ | $72.08 \pm 8.34$ | $56.79 \pm 1.58$ |
| + LA | $94.69 \pm 1.60$ | $77.91 \pm 4.16$ | $55.34 \pm 11.19$ | $80.08 \pm 1.55$ | $70.54 \pm 1.82$ | $55.31 \pm 3.27$ | $77.1 \pm 3.52$ | $70.33 \pm 8.01$ | $56.81 \pm 1.56$ |
| + CAROT | $95.07 \pm 1.20$ | $75.53 \pm 7.42$ | $53.07 \pm 12.99$ | $80.29 \pm 2.33$ | $70.88 \pm 2.22$ | $57.85 \pm 4.05$ | $77.58 \pm 3.04$ | $72.08 \pm 8.34$ | $57.09 \pm 1.90$ |
| LP(ALG1) | $96.09 \pm 0.41$ | $78.34 \pm 4.80$ | $59.91 \pm 6.63$ | $78.56 \pm 1.52$ | $69.71 \pm 0.03$ | $57.61 \pm 3.09$ | $73.39 \pm 9.35$ | $69.28 \pm 11.78$ | $63.67 \pm 7.04$ |
| + LA | $95.81 \pm 0.74$ | $78.97 \pm 4.09$ | $59.98 \pm 6.56$ | $78.48 \pm 1.53$ | $69.71 \pm 0.03$ | $57.47 \pm 3.09$ | $73.39 \pm 9.35$ | $69.21 \pm 11.86$ | $63.67 \pm 7.04$ |
| + CAROT | $96.13 \pm 0.38$ | $80.78 \pm 2.36$ | $59.71 \pm 6.35$ | $78.93 \pm 1.85$ | $70.32 \pm 0.86$ | $57.62 \pm 3.08$ | $73.39 \pm 9.35$ | $74.30 \pm 7.54$ | $64.39 \pm 6.43$ |

| Algorithms | Original $\rho = 0$ | LT $\rho = 5$ | LT $\rho = 15$ | LT $\rho = 50$ | Algorithms | Original $\rho = 0$ | LT $\rho = 5$ | LT $\rho = 15$ | LT $\rho = 50$ |
| --- | --- | --- | --- | --- | --- | --- | --- | --- | --- |
| SL | $69.82 \pm 0.53$ | $67.94 \pm 0.40$ | $69.04 \pm 0.03$ | $74.65 \pm 0.44$ | LP(EMP) | $79.41 \pm 1.33$ | $79.24 \pm 1.03$ | $68.40 \pm 1.90$ | $70.29 \pm 1.62$ |
| + LA | $69.83 \pm 0.53$ | $67.93 \pm 0.41$ | $68.70 \pm 0.30$ | $74.62 \pm 0.36$ | + LA | $79.41 \pm 1.33$ | $79.24 \pm 1.03$ | $68.40 \pm 1.90$ | $70.29 \pm 1.62$ |
| + CAROT | $69.82 \pm 0.53$ | $67.93 \pm 0.41$ | $68.70 \pm 0.41$ | $74.15 \pm 0.47$ | + CAROT | $79.41 \pm 1.33$ | $79.28 \pm 0.91$ | $77.10 \pm 1.74$ | $80.71 \pm 1.50$ |
| RECORDS | $48.71 \pm 3.90$ | $48.15 \pm 4.56$ | $50.14 \pm 1.10$ | $55.12 \pm 1.40$ | LP(ALG1) | $80.23 \pm 0.70$ | $81.27 \pm 0.71$ | $81.99 \pm 0.51$ | $83.44 \pm 0.48$ |
| + LA | $54.12 \pm 2.00$ | $45.48 \pm 2.31$ | $56.83 \pm 1.30$ | $60.87 \pm 1.20$ | + LA | $80.20 \pm 0.74$ | $81.26 \pm 0.72$ | $81.99 \pm 0.51$ | $83.44 \pm 0.48$ |
| + CAROT | $68.16 \pm 0.47$ | $69.04 \pm 0.74$ | $71.70 \pm 0.84$ | $75.69 \pm 0.90$ | + CAROT | $68.90 \pm 11.09$ | $76.38 \pm 5.68$ | $82.00 \pm 0.51$ | $83.44 \pm 0.48$ |

**Conclusions.** We observed many interesting phenomena: (i) training time mitigation can significantly improve the accuracy; (ii) state-of-the-art on training time mitigation might not be appropriate for NESY; (iii) approximate techniques for estimating $\mathbf{r}$ can sometimes be more effective when used for training time mitigation – however, it is robust to softmax reparametrization; (iv) testing time mitigation can substantially improve the accuracy of a classifier; however, it tends to be less effective than training time mitigation; (v) CAROT may be sensitive to the quality of estimated ratios $\hat{\mathbf{r}}$; (vi) Algorithm 1 offers quite accurate marginal estimates.

Starting from the last conclusion, Figure 4 shows that Algorithm 1 offers quite accurate estimates even in challenging

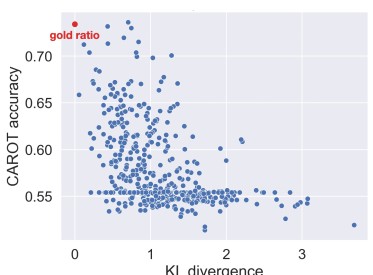

Figure 3: Impact of the label ratio quality on CAROT's performance.

scenarios with high imbalance ratios. Regarding (i), let us focus on Table 1. We can see that both LP(EMP) and LP(ALG1) lead to higher accuracy than models trained exclusively via SL. For example, when $\rho = 5$ in Smallest Parent, the mean accuracy obtained via training under SL is 67.94%; the mean accuracy increases to 79.24% under LP(EMP) and to 81.27% under LP(ALG1). In MAX-4,

the mean accuracy under SL is 55.48%, increasing to 78.56% under LP(ALG1). Regarding (ii), consider again Table 1: when RECORDS is applied jointly with SL, the accuracy of the model can drop substantially, e.g., when $\rho = 5$ in Table 1, the mean accuracy drops from 67.94% to 48.15%. The above stresses the importance of LP (Section 4.2).

Let us move to (iii). In most of the cases, LP(ALG1) leads to higher accuracy than LP(EMP). However, the opposite may also hold in some cases. One such example is MAX-3 for $\rho = 50$: the mean accuracy for the baseline model is 66.74%, increasing to 72.23% under LP(ALG1) and to 77.16% under LP(EMP). The above suggests that there can be cases where employing the gold ratios is not the best solution. A similar observation is made in [18]. One cause of this phenomenon is the high number of classification errors during the initial stages of learning. Those classification errors can become higher in our experimental setting, as in MAX-$M$, we only consider a subset of the pre-images of each weak label to compute SL and (5), to reduce the computational overhead of computing all pre-images. We conclude with CAROT. Table 1 shows that CAROT can be more effective than LA. For example, in Smallest Parent and $\rho = 50$, the mean accuracy of LP(EMP) increases from 70.29% to 80.71% under CAROT; LA has no impact. CAROT may also improve the accuracy of RECORDS models, often, by a large margin. For example, for Smallest Parent and $\rho = 15$, the mean accuracy of a RECORDS-trained model increases from 50.14% to 71.70% when CAROT is applied. CAROT is also consistently better than LA when applied on top of RECORDS. However, there may be cases where LA and CAROT drop the accuracy of the baseline model. One such example is met in Smallest Parent and $\rho = 5$.

We analyze the sensitivity of CAROT to the quality of the input $\hat{\mathbf{r}}$. Quality is measured by means of the KL divergence to $\mathbf{r}$. Figure 3 shows the accuracy of an MNIST model (trained with the MAX-3 dataset), when CAROT is applied at testing time using 500 randomly generated ratios $\hat{\mathbf{r}}$ of varying quality. We observe

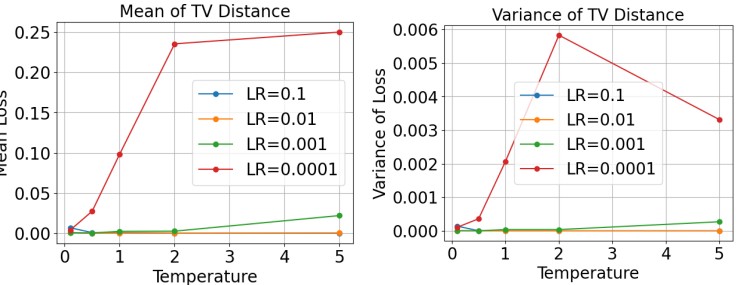

Figure 5: Sensitivity of Algorithm 1 to softmax reparameterization.

that CAROT's effectiveness drops as the estimated marginal diverges more from $\mathbf{r}$. Its performance may also decrease by more than 10% with only a small perturbation in the KL divergence. This instability may be the reason CAROT fails to improve a base model.

To test the sensitivity of CAROT to softmax reparameterization, we performed an additional empirical analysis for MAX-3. We consider a range of different learning rates (LR) ($\{0.1, 0.01, 0.001, 0.0001\}$) and temperatures ($\{0.1, 0.5, 1, 2, 5\}$) when running Algorithm 1. In each run, we randomly generate (1) a true label ratio and (2) 20 initialization points for Algorithm 1. We run the Adam optimizer for 10,000 iterations and compute the total variation (TV) distance between the estimated label ratio and the gold label ratio. Then, we compute the mean and variance of the TV for each experiment. The results are shown in Figure 5. We see that when the temperature is $\leq 2$ and the learning rate $\in \{0.1, 0.01, 0.001\}$ (which are typical choices in machine learning experiments), CAROT consistently achieves $< 0.01$ TV distance, suggesting its robustness.

## 6 Related work

A more detailed comparison against the related work is in Appendix E.

**NESY.** We start with some recent theoretical results and training techniques for NESY. The authors at [67] show PAC learnability for NESY, the authors at [40] characterize the number of deterministic optimal neural classifiers as a function of $\sigma$ and propose techniques to improve learning accuracy. However, they make additional assumptions about the training data or the classifiers. In contrast, we propose imbalance mitigation techniques without making additional assumptions. The authors in [28] and [39] propose learning techniques based on unified expectation maximization [54] and entropy regularization, respectively. Unlike our work, none of the above studies empirically or theoretically learning imbalances in NESY. An interesting direction is to combine the active learning strategy in [39] with our training time mitigation technique. In particular, we could encourage the acquisition of labels for classes that maximize the entropy and, at the same time, appear with smaller ratios. The

latter can be achieved, for example, by assigning a higher weight to classes with smaller (estimated) ratios in the entropy computation.

**Long-tailed supervised learning**. Two supervised learning techniques related to our work are LA [42] and OTLM [46]. Both aim at testing time mitigation. LA modifies the classifier's scores by subtracting the gold ratios. CAROT can be substantially more effective than LA, see Section 5. OTLM assumes that the marginal $\mathbf{r}$ is known, resorting to an OT formulation to adjust the classifier's scores. In contrast, we propose a statistically consistent technique to estimate $\mathbf{r}$, see Section 4.1, and resort to RSOT to accommodate noisy $\widehat{\mathbf{r}}$'s. Finally, well-known re-weighting schemes [2, 53] are not applicable to our setting: they require access to the gold labels; we assume the gold labels are hidden.

**Long-tailed PLL**. There is no previous work on long-tailed MI-PLL. Hence, we focus on standard PLL. The authors in [10] showed that certain classes are harder to learn than others in PLL. We are the first to extend these results to NeSy. The only two works at the intersection of long-tailed learning and PLL are RECORDS [18] and SoLar [65]. RECORDS modifies the classifier's scores using the same idea as LA and employs a momentum-updated prototype feature to estimate $\widehat{\mathbf{r}}$. Section 5 shows that RECORDS is less effective than our proposals, degrading the baseline accuracy on multiple occasions. SoLar cannot act as a competitor to our technique, since it cannot be straightforwardly extended to handle training samples with multiple instances, see Appendix E.

## 7 Conclusions and Future Work

**Comments on the theory.** In Section 3, the probability of misclassifying an instance $x$ depends only on its class. This assumption is also adopted in other settings, such as *noisy label learning* [74, 45]. Although there are scenarios where this assumption does not hold, our theory is an over-approximation to those scenarios similarly to the connection between class- and instance-dependent noisy label learning. Our analysis in Section 3 assumes that the weak label $s$ depends on the instance $x$ only via its class. Nevertheless, it is straightforward to extend our theory to instance-dependent symbolic components $\sigma$. This can be done by partitioning the input space into sub-regions, where in each region, the symbolic component is a fixed function. Generalization bounds can then be derived per region using our methods and averaged to obtain an overall bound. Furthermore, our formulation in (2) can be extended when the instances $(x_1, \ldots, x_M)$ have few correlations. Our theory is a good starting point for cases where these correlations are strong, since learning imbalances will also occur in these cases, but now are easier to describe.

**Training vs testing time mitigation.** CAROT is a more lightweight technique, however, it may lead to lower classification accuracy than LP. On the contrary, LP *may* increase the training overhead over the state-of-the-art, namely training by applying the top-$k$ SL per training sample [71, 67]. This is because when $k$ is fixed, the complexity of computing the SL is polynomial; in contrast, solving (5), which is a linear program calculated out of a batch of samples, is an NP-hard problem. However, when the SL runs *without* approximations and the pre-image of $\sigma$ is very large, the complexity of SL is worst case #P-complete per training sample [8], making (5) more computationally efficient. From a computational viewpoint, it is worth stressing that the LP has a linear growth in $\sigma^{-1}$ as we employ the Tseytin transformation D to translate from the pre-image into the ILP. The complexity of enumerating $\sigma^{-1}$ is inherent to all relevant NeSy frameworks [37, 20, 61], as they all rely on the pre-image to compute a loss, see the discussion in [67]. To reduce the computational overhead of our ILP-based technique, we could adopt multiple techniques to solve ILP efficiently [22]. We could treat the program in (5) as a differentiable layer by applying optimization techniques as in [1]. When allowing the variables to take values in $[0, 1]$, (5) becomes an LP. Then, we can employ the simplex algorithm, which runs quite efficiently in practice [56].

We are the first to theoretically characterize and mitigate learning imbalances in NeSy. Our characterization complements the existing theory in long-tailed learning, identifying and addressing the unique challenges in NeSy. Our empirical analysis revealed two topics for future research: *computing marginals for testing time mitigation* and *designing more effective testing time mitigation techniques*.

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

## Appendix Organization

Our appendix is organized as follows:

- Appendix A introduces notions related to (robust) optimal transport and discusses the relationship between our notation and the notation used in the relevant NSL literature.
- Appendix B provides the proofs of all the formal statements in Section 3 and a more detailed discussion of our error bounds.
- Appendix C provides the proof of statistical consistency of Algorithm 1 and discusses other technical aspects related to Algorithm 1.
- Appendix D discusses a nonlinear program formulation of NESY. In addition, it presents in detail the steps to derive the optimization objective in (5), as well as an example of (5) in the context of Example 1.1.
- Appendix E presents an extended version of the related work.
- Appendix F provides further details on our empirical analysis and presents results on more benchmarks.
- Tables 5 and 6 summarize our notation.

## A  Extended Preliminaries

**Optimal transport.** Let $Z_1$ and $Z_2$ be two discrete random variables over $[m_1]$ and $[m_2]$. For $i \in [2]$, vector $\mathbf{b}^i \in \mathbb{R}_+^{m_i}$ denotes the probability distribution of $Z_i$, i.e., $\mathbb{P}(Z_i = m_j) = b_j^i$, for each $j \in [m_i]$. Let $U$ be the set of matrices defined as $\{\mathbf{Q} \in \mathbb{R}_+^{m_1 \times m_2} | \mathbf{Q}\mathbf{1}_{m_1} = \mathbf{b}^2, \mathbf{Q}\mathbf{1}_{m_2} = \mathbf{b}^1\}$. The *optimal transport* (OT) problem [48] asks us to find the matrix $\mathbf{Q} \in U$ that maximizes a linear object subject to marginal constraints, namely

$$\min_{\mathbf{Q} \in U} \langle \mathbf{P}, \mathbf{Q} \rangle \tag{7}$$

Assume that we are strict in enforcing the probability distribution $\mathbf{b}^1$, but not in enforcing $\mathbf{b}^2$. The *robust semi-constrained optimal transport* (RSOT) problem [26] aims to find:

$$\min_{\mathbf{Q} \in U'} \langle \mathbf{P}, \mathbf{Q} \rangle + \tau \mathrm{KL}(\mathbf{Q}\mathbf{1}_{m_1} || \mathbf{b}^2) \tag{8}$$

where $U' = \{\mathbf{Q} \in \mathbb{R}_+^{m_1 \times m_2} | \mathbf{Q}\mathbf{1}_{m_2} = \mathbf{b}^1\}$ and $\tau > 0$ is a regularization parameter. The solution to (8) can be approximated in polynomial time using *robust semi-Sinkhorn* from [26], which generalizes the classical Sinkhorn algorithm [11] for OT.

**Other NESY notation.** We now show that our notation for NESY samples is equivalent to the notation adopted by previous works on the topic [37, 20, 40].

Let $\mathcal{K}$ be a background logical theory that "sits" on top of $f$, i.e., it reasons over the predictions of $f$. In practice, we may have one or more classifiers, $f_1, \ldots, f_N$, each with its own input and output domains. To simplify the description, we focus on the single-classifier case. However, both our notation and the notation in [37, 20, 40] can be trivially extended to support these scenarios.

In previous works, NESY training samples may be denoted by $(\mathbf{x}, \phi)$, where $\mathbf{x}$ is a set of elements from $\mathcal{X}$ and $\phi$ is a logical sentence (or a single target fact in the simplest scenario). The gold labels of the input instances are unknown to the learner. Instead, we only know that the gold labels of the elements in $\mathbf{x}$ satisfy the logical sentence $\phi$ subject to $\mathcal{K}$. The sentence $\phi$ and the logical theory $\mathcal{K}$ allow us to "guess" what the gold labels of the elements in $\mathbf{x}$ might be so that $\phi$ is logically satisfied subject to $\mathcal{K}$. This is essentially the process of *abduction* [61]. To align with the terminology in our paper, for a training sample $(\mathbf{x}, \phi)$, we use the term *pre-image*[2] to denote a combination of labels of the elements in $\mathbf{x}$, such that $\phi$ is logically satisfied subject to $\mathcal{K}$. The gold pre-image is the one mapping each instance to its gold label. Abduction allows us to "get rid of" $\phi$ and $\mathcal{K}$ and represent each training sample via $\mathbf{x}$ and its corresponding pre-images, i.e., as $(\mathbf{x}, \{\sigma_i\}_{i=1}^{\omega})$, where each pre-image $\sigma_i$ is a mapping from $\mathbf{x}$ into labels in $\mathcal{Y}$. By assuming a canonical ordering on the elements in $\mathbf{x}$, we can view each $\sigma_i(\mathbf{x})$ as a vector of labels, one for each element in $\mathbf{x}$. Therefore,

---

[2]Pre-images correspond to *proofs* in [61, 20, 12, 37].

we can equivalently see each training sample as a tuple of the form $(\mathbf{x}, \{\sigma_i(\mathbf{x})\}_{i=1}^{\omega})$, supporting our claim that the two notations are equivalent.

# B   Proofs and Details on Section 3

## B.1   Proofs

**Proposition 3.1** (Class-specific risk bound). *For any $j \in \mathcal{Y}$, we have that $R_j(f) \leq \Phi_{\sigma,j}(R_\mathsf{P}(f; \sigma))$.*

*Proof.* This result follows directly from the definition of the program (2). $\qquad\square$

**Proposition 3.3.** *Let $d_{[\mathcal{F}]}$ be the Natarajan dimension of $[\mathcal{F}]$. Given a confidence level $\delta \in (0, 1)$, we have that $R_j(f) \leq \Phi_{\sigma,j}(\widetilde{R}_\mathsf{P}(f; \sigma, \mathcal{T}_\mathsf{P}, \delta))$ with probability $1 - \delta$ for any $j \in [c]$, where*

$$\widetilde{R}_\mathsf{P}(f; \sigma, \mathcal{T}_\mathsf{P}, \delta) = \widehat{R}_\mathsf{P}(f; \sigma, \mathcal{T}_\mathsf{P}) + \sqrt{\frac{2\log(em_\mathsf{P}/2d_{[\mathcal{F}]}\log(6Mc^2 d_{[\mathcal{F}]}/e))}{m_\mathsf{P}/2d_{[\mathcal{F}]}\log(6Mc^2 d_{[\mathcal{F}]}/e)}} + \sqrt{\frac{\log(1/\delta)}{2m_\mathsf{P}}} \qquad (3)$$

*Proof.* Let $L_\sigma \circ [\mathcal{F}]$ be the function space that maps a (training) example $(\mathbf{x}, s)$ to its partial loss defined as follows:

$$L_\sigma \circ [\mathcal{F}] := \{(\mathbf{x}, s) \mapsto L_\sigma([f](\mathbf{x}), s) | f \in \mathcal{F}\} \qquad (9)$$

The standard generalization bound with VC dimension (see, for example, Corollary 3.19 of [44]) implies that:

$$R_\mathsf{P}(f) \leq \widehat{R}_\mathsf{P}(f; \mathcal{T}_\mathsf{P}) + \sqrt{\frac{2\log(em_\mathsf{P}/d_{\mathrm{VC}}(L_\sigma \circ [\mathcal{F}]))}{m_\mathsf{P}/d_{\mathrm{VC}}(L_\sigma \circ [\mathcal{F}])}} + \sqrt{\frac{\log(1/\delta)}{2m_\mathsf{P}}} \qquad (10)$$

where $d_{\mathrm{VC}}(\cdot)$ is the VC dimension. For simplicity, let $d = d_{\mathrm{VC}}(L_\sigma \circ [\mathcal{F}])$ and $d_{[\mathcal{F}]}$ be the Natarajan dimension of $[\mathcal{F}]$. Using a similar argument as in [67], given any $d$ samples in $\mathcal{X}^M \times \mathcal{O}$ using $[\mathcal{F}]$, we let $N$ be the maximum number of distinct ways to assign label vectors (in $\mathcal{Y}^M$) to these $d$ samples. Then, the definition of VC-dimension implies that:

$$2^d \leq N \qquad (11)$$

On the other hand, these $d$ samples contain $Md$ input instances in $\mathcal{X}$. By Natarajan's lemma (see, for example, Lemma 29.4 of [55]), we have that:

$$N \leq (Md)^{d_{[\mathcal{F}]}} c^{2d_{[\mathcal{F}]}} \qquad (12)$$

Combining (12) with the above equations, it follows that

$$(Md)^{d_{[\mathcal{F}]}} c^{2d_{[\mathcal{F}]}} \geq N \geq 2^d \qquad (13)$$

Taking the logarithm on both sides, we have that:

$$d_{[\mathcal{F}]} \log(Md) + 2d_{[\mathcal{F}]} \log c \geq d \log 2 \qquad (14)$$

Taking the first-order Taylor series expansion of the logarithm function at the point $6d_{[\mathcal{F}]}$, we have:

$$\log(d) \leq \frac{d}{6d_{[\mathcal{F}]}} + \log(6d_{[\mathcal{F}]}) - 1 \qquad (15)$$

Therefore,

$$\begin{aligned} d \log 2 &\leq d_{[\mathcal{F}]} \log d + d_{[\mathcal{F}]} \log M + 2d_{[\mathcal{F}]} \log c \\ &\leq d_{[\mathcal{F}]} \left( \frac{d}{6d_{[\mathcal{F}]}} + \log(6d_{[\mathcal{F}]}) - 1 \right) + d_{[\mathcal{F}]} \log M + 2d_{[\mathcal{F}]} \log c \\ &= \frac{d}{6} + d_{[\mathcal{F}]} \log(6Mc^2 d_{[\mathcal{F}]}/e) \end{aligned} \qquad (16)$$

Rearranging the inequality yields

$$\begin{aligned} d &\leq \frac{d_{[\mathcal{F}]} \log(6Mc^2 d_{[\mathcal{F}]}/e)}{\log 2 - 1/6} \\ &\leq 2d_{[\mathcal{F}]} \log(6Mc^2 d_{[\mathcal{F}]}/e) \end{aligned} \qquad (17)$$

as claimed. $\qquad\square$

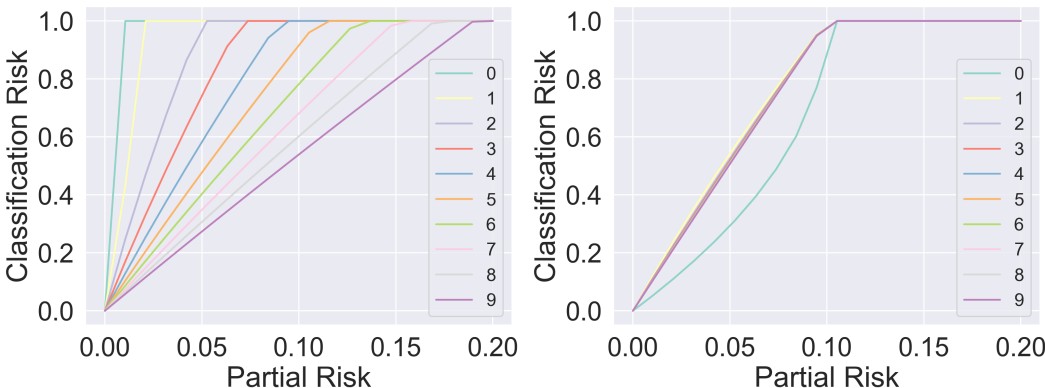

Figure 6: Class-specific upper bounds obtained via (2). (left) $\mathcal{D}_Y$ is uniform. (right) $\mathcal{D}_{\mathsf{P}_S}$ is uniform. (Enlarged version of Figure 2).

**Proposition 3.4.** *If $\sigma$ is $M$-unambiguous, we have*

$$R(f) \leq \sqrt{\mathbf{w}^{\mathsf{T}}(D(\boldsymbol{\Sigma}_{\sigma,\mathbf{r}}))^{\dagger}\mathbf{w}R_{\mathsf{P}}(f;\sigma)} = \sqrt{c(c-1)R_{\mathsf{P}}(f;\sigma)} \tag{4}$$

*which coincides with Lemma 1 from [67] for $M = 2$, where $\mathbf{w} := \sum_{j=1}^{c} r_j \mathbf{w}_j$.*

*Proof.* Since $\mathbf{w} := \sum_{i=1}^{c} r_i \mathbf{w}_i$, we have $R(f) = \mathbf{w}^{\mathsf{T}}\mathbf{h}$. Then, we consider the following relaxed program:

$$\begin{aligned} \max_{\mathbf{h}} \quad & \mathbf{w}^{\mathsf{T}}\mathbf{h} \\ \text{s.t.} \quad & \mathbf{h}^{\mathsf{T}}D(\boldsymbol{\Sigma}_{\sigma,\mathbf{r}})\mathbf{h} \leq R_{\mathsf{P}} \end{aligned} \tag{18}$$

where $D(\boldsymbol{\Sigma}_{\sigma,\mathbf{r}})$ is the diagonal part of $\boldsymbol{\Sigma}_{\sigma,\mathbf{r}}$, namely:

$$D(\boldsymbol{\Sigma}_{\sigma,\mathbf{r}}) = [r_i r_j \mathbb{1}\{i = j\}\mathbb{1}\{i \not\equiv j \ (\mathrm{mod}\ c)\}]_{i \in [c^2], j \in [c^2]} \tag{19}$$

In other words, $D(\boldsymbol{\Sigma}_{\sigma,\mathbf{r}})$ encodes all the partial risks caused by repeating the same type of misclassification twice. On the other hand, the $M$-unambiguity condition ensures that each type of misclassification, when repeated twice, leads to a misclassification of the weak label. Therefore, $\mathbf{w} \in \mathrm{Range}(D(\boldsymbol{\Sigma}_{\sigma,\mathbf{r}}))$.

The problem in (18) is a special case of the single-constraint quadratic optimization problem. Then, the fact that $\mathbf{w} \in \mathrm{Range}(D(\boldsymbol{\Sigma}_{\sigma,\mathbf{r}}))$ implies that the dual function of this problem (with dual variable $\lambda$) is

$$g(\lambda) = \lambda R_{\mathsf{P}} + \frac{\mathbf{w}^{\mathsf{T}}(D(\boldsymbol{\Sigma}_{\sigma,\mathbf{r}}))^{\dagger}\mathbf{w}}{4\lambda} \tag{20}$$

where $(D(\boldsymbol{\Sigma}_{\sigma,\mathbf{r}}))^{\dagger}$ is the pseudo-inverse, namely

$$(D(\boldsymbol{\Sigma}_{\sigma,\mathbf{r}}))^{\dagger} = [(r_i r_j)^{-1}\mathbb{1}\{i = j\}\mathbb{1}\{i \not\equiv j \ (\mathrm{mod}\ c)\}]_{i \in [c^2], j \in [c^2]} \tag{21}$$

Therefore,

$$\mathbf{w}^{\mathsf{T}}(D(\boldsymbol{\Sigma}_{\sigma,\mathbf{r}}))^{\dagger}\mathbf{w} = c(c-1) \tag{22}$$

According to Appendix B of [3], strong duality holds for this problem. Therefore, the optimal value is given exactly as

$$\inf_{\lambda \geq 0} g(\lambda) = 2\sqrt{\frac{c(c-1)}{4}R_{\mathsf{P}}} = \sqrt{c(c-1)R_{\mathsf{P}}} \tag{23}$$

as claimed. $\qquad\square$

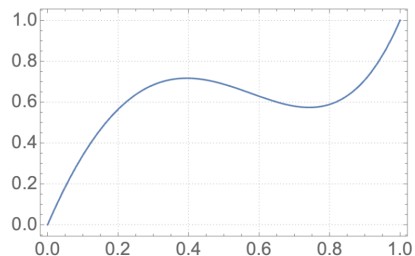

Figure 7: Plot of function $t \mapsto t^4 + 6t^2(1-t)^2 + 4t(1-t)^3$.

## B.2 Further Discussion on the Proposed Risk Bounds

Intuitively, the difficulty of learning is affected by (i) the distribution of weak labels in $\mathbf{D}_\mathsf{P}$ and (ii) the size of the pre-image of $\sigma$ for each weak label. These two factors are reflected in our risk-specific bounds. Let us continue with the analysis in Example 3.2.

**Example B.1** (Cont' Example 3.2). *Let us start with* CASE 1. *In this case, our class-specific error bounds suggest that learning the zero class is more difficult than learning nine class, despite the fact that both hidden labels $y_1$ and $y_2$ are uniform in $\{0, \ldots, 9\}$, see the left side of Figure B.2. The root cause of this learning imbalance is $\sigma$ and its characteristics. In particular, the weak labels that result after independently drawing pairs of MNIST digits and applying $\sigma$ on their gold labels are long-tailed, with $s = 0$ occurring with probability $1/100$ and $s = 9$ occurring with probability $17/100$ in the training data. Hence, we have more supervision to learn class nine than to learn zero.*

*Now, let us focus on* CASE 2. *In this case, our class-specific bounds suggest that learning class zero is the easiest to learn – see right side of Figure B.2. This is due to two reasons. First, the weak labels follow the same uniform distribution. Hence, we have the same amount of supervision to learn all classes. Second, the pre-image of $\sigma$ for different weak labels is very different. Regarding the second reason, the weak label $s = 0$ provides much stronger supervision than the weak label $s = 9$: when $s = 0$, we have direct supervision ($s = 0$ implies $y_1 = y_2 = 0$); in contrast, when $s = 9$ this only means that $y_1 = 9$ and $y_2$ is any label in $\{0, \ldots, 9\}$, or vice versa.*

The above shows that $\sigma$ (i) can lead to imbalances in the weak labels even if the hidden labels are uniformly distributed and (ii) may provide supervision signals of very different strengths. Hence, learning in NESY is *inherently imbalanced* due to $\sigma$.

## B.3 Details on Plotting Figure 2

In this subsection, we describe the steps we followed to create the plots in Figure 2. We produced the curves shown in each figure by plotting 20 evenly spaced points within the partial risk interval $R_\mathsf{P} \in [0, 0.2]$. To obtain the value of the classification risk at each point, we solved the optimization program (2) using the COBYLA optimization algorithm implemented by the `scipy.optimize` package. To mitigate numerical instability, for each point, we ran the optimization solver ten times and then dropped invalid results that were not in the range $[0, 1]$. The median of the remaining valid results was then taken as the solution to (2).

## C Further details on Algorithm 1

**Proof of statistical consistency of Algorithm1.** The approximation $\widehat{\mathbf{r}}$ given by Algorithm 1 can be viewed as a method to find the maximum likelihood estimation whose consistency is guaranteed under suitable conditions. The most critical is the invertibility of $\Psi_\sigma$. The invertibility is satisfied by practical transitions as the one in Example 1.1, but may fail to hold for certain transitions even if the $M$-unambiguity condition [67] holds. We will provide one such example later in this section.

Suppose that the backprobagation step in Algorithm 1 can find the maximum likelihood estimator. For a real $\epsilon > 0$, let $\Delta_c^\epsilon$ be the shrinked probability simplex defined as $\Delta_c^\epsilon := \{\mathbf{r} \in \Delta_c | r_j \geq \epsilon \, \forall j \in [c]\}$. Let $\widehat{\mathbf{r}}_{m_\mathsf{p}}^* := \mathrm{argmin}_{\widehat{\mathbf{r}} \in \Delta_c^\epsilon} \sum_{j=1}^{c_S} \bar{p}_j \log[\Psi_\sigma(\widehat{\mathbf{r}})]_j$ be the maximum likelihood estimation. We have:

**Proposition C.1** (Consistency). *If there exists an $\epsilon > 0$, such that $\mathbf{r} \in \Delta_c^\epsilon$ and $\Psi_\sigma$ is injective in $\Delta_c^\epsilon$, then $\widehat{\mathbf{r}}_{m_\mathsf{P}}^* \to \mathbf{r}$ in probability as $m_\mathsf{P} \to \infty$.*

*Proof.* Let $\Delta_{c_S}^{\sigma,\epsilon} := \{\Psi_\sigma(\mathbf{r}) | \mathbf{r} \in \Delta_c^\epsilon\}$ be the image of $\Psi_\sigma$ on $\Delta_c^\epsilon$. The set $\Delta_{c_S}^{\sigma,\epsilon}$ is a compact subset in $\mathbb{R}^{c_S}$. For any weak label $a_j \in \mathcal{S}$, let $H(a_j, \mathbf{r}) := -\log([\Psi_\sigma(\mathbf{r})]_j)$ be the point-wise log-likelihood. The $M$-unambiguity condition ensures that each coordinate of every vector in $\Delta_{c_S}^{\sigma,\epsilon}$ should be at least $\epsilon^M$, and hence the function $H$ is bounded on $\Delta_{c_S}^{\sigma,\epsilon}$. By Theorem 1 of [23], this ensures that $\sum_s H(s, \mathbf{r})$ converges uniformly to $\mathbb{E}_S[H(S, \mathbf{r})]$. According to [64] (Theorem 5.7), the uniform convergence further ensures that $\Psi_\sigma(\widehat{\mathbf{r}}_{m_\mathsf{P}}^*) \to \mathbf{p}$ in probability as $m_\mathsf{P} \to \infty$. Since $\Psi_\sigma$ is invertible, this implies that $\widehat{\mathbf{r}}_{m_\mathsf{P}}^* \to \mathbf{r}$ in probability. $\qquad\square$

**Counterexample where the invertibility of $\Psi_\sigma$ does not hold.** Consider the following transition function for binary labels ($\mathcal{Y} = \{0, 1\}$) and $M = 4$:

$$\sigma(y_1, y_2, y_3, y_4) = \begin{cases} 1, & \sum_{i=1}^{4} y_i \in \{1, 2, 4\} \\ 0, & \text{otherwise} \end{cases} \tag{24}$$

The $M$-unambiguity condition [67] holds since $\sigma(0, 0, 0, 0) \neq \sigma(1, 1, 1, 1)$. On the other hand, the probability the weak label is one can be expressed as:

$$\mathbb{P}(s = 1) = r_1^4 + 6r_1^2 r_0^2 + 4r_1 r_0^3 = r_1^4 + 6r_1^2(1 - r_1)^2 + 4r_1(1 - r_1)^3 \tag{25}$$

which is not an injection, see the plot of function $t \mapsto t^4 + 6t^2(1 - t)^2 + 4t(1 - t)^3$ in Figure 7.

# D    Details on Section 4.2

## D.1    A Nonlinear Program Formulation

A straightforward idea that accommodates the requirements set in Section 4.2 is to reformulate (8) by (i) extending $\mathbf{P}$ (resp. $\mathbf{Q}$) to a tensor of size $n \times c \times M$ to store the scores (resp. pseudolabels) of $M$-ary tuples of instances and (ii) modifying $U'$ so that the product of the combinations of entries in $\mathbf{Q}$ corresponding to invalid label assignments is forced to zero. However, modifying $U'$ in this way, we cannot employ Sinkhorn-like techniques as the one in [31], leaving us only with the option to employ nonlinear[3] programming techniques to find $\mathbf{Q}$.

## D.2    Deriving the Linear Program in (5)

Let $(x_{\ell,1}, \ldots, x_{\ell,M}, s_\ell)$ denote the $\ell$-th NESY training sample, where $\ell \in [n]$. To derive the linear program in (5), we associate each weak label $s_\ell$ with a DNF formula $\Phi_\ell$, a process that is standard in the neurosymbolic literature [71, 61, 20, 67]. To ease the presentation, we describe how to compute $\Phi_\ell$. Let $\{\mathbf{y}_{\ell,1}, \ldots, \mathbf{y}_{\ell,R_\ell}\}$ be the set of vectors of labels in $\sigma^{-1}(s_\ell)$. We associate each prediction with a Boolean variable. Namely, let $q_{\ell,i,j}$ be a Boolean variable that becomes true when $x_{\ell,i}$ is assigned with label $j \in \mathcal{Y}$. Via associating predictions with Boolean variables, each $\mathbf{y}_{\ell,t}$ can be associated with a conjunction $\varphi_{\ell,t}$ over Boolean variables from $\{q_{\ell,i,j} | i \in [M], j \in [c]\}$. In particular, $q_{\ell,i,j}$ occurs in $\phi_{\ell,t}$ only if the $i$-th label in $\mathbf{y}_{\ell,t}$ is $j \in \mathcal{Y}$. Consequently, the training sample $(x_{\ell,1}, \ldots, x_{\ell,M}, s_\ell)$ is associated with the DNF formula $\Phi_\ell = \bigvee_{r=1}^{R_\ell} \varphi_{\ell,t}$ that encodes all vectors of labels in $\sigma^{-1}(s_\ell)$. We assume a canonical ordering over the variables occurring in $\varphi_{\ell,t}$, using $\varphi_{\ell,t,j}$ to refer to the $j$-th variable, and use $|\varphi_{\ell,t}|$ to denote the number of (unique) Boolean variables occurring $\varphi_{\ell,t}$. Based on the above, we have $\varphi_{\ell,t} = \bigwedge_{k=1}^{|\varphi_{\ell,t}|} \varphi_{\ell,t,k}$.

Similarly to [57], we use the Iverson bracket $[]$ to map Boolean variables to their corresponding integer ones, e.g., $[q_{\ell,i,j}]$, denotes the integer variable associated with the Boolean variable $q_{\ell,i,j}$.

We are now ready to construct linear program (5). Notice that the solutions of this program capture the label assignments that abide by $\sigma$, i.e., the labels assigned to each $(x_{\ell,1}, \ldots, x_{\ell,M})$ should be either of $\mathbf{y}_{\ell,1}, \ldots, \mathbf{y}_{\ell,R_\ell}$. The steps of the construction are (see [57]):

---

[3]Nonlinearity comes from the KL term and by enforcing invalid label combinations to have product equal to zero.

- (STEP 1) We translate each $\Phi_\ell$ into a CNF formula $\Phi'_\ell$ via the Tseytin transformation [62] to avoid the exponential blow up of the (brute force) DNF to CNF conversion.
- (STEP 2) We add the corresponding linear constraints out of each subformula in $\Phi'_\ell$.

Given $\Phi_\ell = \bigvee_{r=1}^{R_\ell} \varphi_{\ell,t}$, the Tseytin transformation associates a fresh Boolean variable $\alpha_{\ell,t}$ with each disjunction $\varphi_{\ell,t}$ in $\Phi_\ell$ and rewrites $\Phi_\ell$ into the following logically equivalent formula:

$$\Phi'_\ell := \underbrace{\bigvee_{t=1}^{R_\ell} \alpha_{\ell,t}}_{\Psi_\ell} \wedge \bigwedge_{t=1}^{R_\ell} (\alpha_{\ell,t} \leftrightarrow \varphi_{\ell,t}) \tag{26}$$

After obtaining $\Phi'_\ell$, the construction of (5) proceeds as follows. The first inequality that will be added to (5) comes from formula $\Psi_\ell$. In particular, it will be the inequality $\sum_{t=1}^{R_\ell} [\alpha_{\ell,t}] \geq 1$, due to Constraint (3) from [57]. The next inequalities come from the subformula $\bigwedge_{t=1}^{R_\ell} (\alpha_{\ell,t} \leftrightarrow \varphi_{\ell,t})$ from (26). The latter can be rewritten to the following two formulas:

$$\alpha_{\ell,t} \to \bigwedge_{k=1}^{|\varphi_{\ell,t}|} \varphi_{\ell,t,k} \tag{27}$$

$$\bigwedge_{k=1}^{|\varphi_{\ell,t}|} \varphi_{\ell,t,k} \to \alpha_{\ell,t} \tag{28}$$

According to Constraint (10) from [57], (27) and (28) are associated with the following inequalities:

$$-|\varphi_{\ell,t}|[\alpha_{\ell,t}] + \sum_{k=1}^{|\varphi_{\ell,t}|} [\varphi_{\ell,t,k}] \geq 0 \tag{29}$$

$$-\sum_{k=1}^{|\varphi_{\ell,t}|} [\varphi_{\ell,t,k}] + [\alpha_{\ell,t}] \geq (1 - |\varphi_{\ell,t}|) \tag{30}$$

which will also be added to the linear program.

Lastly, according to Constraint (5) from [57], we have an equality $\sum_{j=1}^c [q_{\ell,i,j}] = 1$, for each $\ell \in [n]$ and $i \in [M]$. The above equality essentially requires the scores of all pseudolabels for a given instance $x_{\ell,i}$ to sum up to one. Finally, we require each pseudolabel $[q_{\ell,i,j}]$ to be in $[0,1]$, for each $\ell \in [n]$, $i \in [M]$, and $j \in [c]$.

Putting everything together, we have the following linear program:

$$
\begin{aligned}
\textbf{minimize} \quad & \min_{(\mathbf{Q}_1,\ldots,\mathbf{Q}_m)} \sum_{i=1}^{M} \langle \mathbf{Q}_i, -\log(\mathbf{P}_i) \rangle, \\
\textbf{subject to} \quad &
\begin{array}{rll}
\sum_{r=1}^{R_\ell} [\alpha_{\ell,t}] & \geq 1, & \ell \in [n], \\
-|\varphi_{\ell,t}|[\alpha_{\ell,t}] + \sum_{k=1}^{|\varphi_{\ell,t}|} [\varphi_{\ell,t,k}] & \geq 0, & \ell \in [n], t \in [R_\ell] \\
-\sum_{k=1}^{|\varphi_{\ell,t}|} [\varphi_{\ell,t,k}] + [\alpha_{\ell,t}] & \geq -1(1 - |\varphi_{\ell,t}|), & \ell \in [n], t \in [R_\ell] \\
\sum_{j=1}^{c} [q_{\ell,i,j}] & = 1, & \ell \in [n], i \in [M] \\
[q_{\ell,i,j}] & \in [0,1], & \ell \in [n], i \in [M], j \in [c]
\end{array}
\end{aligned}
\tag{31}
$$

Program (5) results after adding to the above program constraints enforcing the hidden label ratios $\hat{\mathbf{r}}$.

**Example D.1.** *We demonstrate an example of* (5) *in the context of Example 1.1. We assume $n = 2$. We also assume that the weak labels $s_1$ and $s_2$ of the two NESY samples in the batch are equal to $0$ and $1$, respectively. Due to the properties of the max, we have:*

$$\sigma^{-1}(0) = \{(0,0)\} \tag{32}$$

$$\sigma^{-1}(1) = \{(0,1), (1,0), (1,1)\} \tag{33}$$

*and formulas $\Phi_1$ and $\Phi_2$ are defined as:*

$$\Phi_1 = \underbrace{q_{1,1,0} \wedge q_{1,2,0}}_{\varphi_{1,1}} \tag{34}$$

$$\Phi_2 = \underbrace{q_{2,1,0} \wedge q_{2,2,1}}_{\varphi_{2,1}} \vee \underbrace{q_{2,1,1} \wedge q_{2,2,0}}_{\varphi_{2,2}} \vee \underbrace{q_{2,1,1} \wedge q_{2,2,1}}_{\varphi_{2,3}} \tag{35}$$

*The Tseytin transformation associates the fresh Boolean variables $\alpha_{1,1}$, $\alpha_{2,1}$, $\alpha_{2,2}$, and $\alpha_{2,3}$ to $\varphi_{1,1}$, $\varphi_{2,1}$, $\varphi_{2,2}$, and $\varphi_{2,3}$, respectively, and rewrites $\Phi_1$ and $\Phi_2$ to the following logically equivalent formulas:*

$$\Phi'_1 = \alpha_{1,1} \wedge (\alpha_{1,1} \leftrightarrow \varphi_{1,1}) \tag{36}$$

$$\Phi'_2 = (\alpha_{2,1} \vee \alpha_{2,2} \vee \alpha_{2,3}) \wedge (\alpha_{2,1} \leftrightarrow \varphi_{2,1}) \wedge (\alpha_{2,2} \leftrightarrow \varphi_{2,2}) \wedge (\alpha_{2,3} \leftrightarrow \varphi_{2,3}) \tag{37}$$

*The linear constraints that are added due to $\Phi'_1$ are:*

$$\begin{array}{rl} [\alpha_{1,1}] & \geq 1 \\ -|\varphi_{1,1}|[\alpha_{1,1}] + [q_{1,1,0}] + [q_{1,2,0}] & \geq 0 \\ -([q_{1,1,0}] + [q_{1,2,0}]) + [\alpha_{1,1}] & \geq -1(1 - |\varphi_{1,1}|) \end{array} \tag{38}$$

*The linear constraints that are added due to $\Phi'_2$ are:*

$$\begin{array}{rl} [\alpha_{2,1}] + [\alpha_{2,2}] + [\alpha_{2,3}] & \geq 1 \\ -|\varphi_{2,1}|[\alpha_{2,1}] + [q_{2,1,0}] + [q_{2,2,1}] & \geq 0 \\ -|\varphi_{2,2}|[\alpha_{2,2}] + [q_{2,1,1}] + [q_{2,2,0}] & \geq 0 \\ -|\varphi_{2,3}|[\alpha_{2,3}] + [q_{2,1,1}] + [q_{2,2,1}] & \geq 0 \\ -([q_{2,1,0}] + [q_{2,2,1}]) + [\alpha_{2,1}] & \geq -1(1 - |\varphi_{2,1}|) \\ -([q_{2,1,1}] + [q_{2,2,0}]) + [\alpha_{2,2}] & \geq -1(1 - |\varphi_{2,2}|) \\ -([q_{2,1,1}] + [q_{2,2,1}]) + [\alpha_{2,3}] & \geq -1(1 - |\varphi_{2,3}|) \end{array} \tag{39}$$

*Finally, the requirement that the pseudolabels for each instance $x_{\ell,i}$ to sum up to one, for $\ell \in [2]$ and $i \in [2]$, and to lie in $[0, 1]$ introduces the following linear constraints:*

$$\begin{array}{rl} \sum_{j=0}^{9}[q_{1,1,j}] & = 1 \\ \sum_{j=0}^{9}[q_{1,2,j}] & = 1 \\ \sum_{j=0}^{9}[q_{2,1,j}] & = 1 \\ \sum_{j=0}^{9}[q_{2,2,j}] & = 1 \\ [q_{1,i,j}] & \in [0,1], \quad i \in [2], j \in \{0, \ldots, 9\} \\ [q_{2,i,j}] & \in [0,1], \quad i \in [2], j \in \{0, \ldots, 9\} \end{array} \tag{40}$$

## E  Extended Related Work

**NESY**. NESY quite often arises in NSL [36, 69, 12, 72, 61, 38, 20, 28, 21]. However, we are the first to study the phenomenon of learning imbalances. Below we discuss some recent theoretical results [40, 39, 67]. The work in [40, 39] deals with the problem of characterizing and mitigating *reasoning shortcuts*. Intuitively, a reasoning shortcut is a classifier that has small partial risk but high classification risk. For example, a reasoning shortcut is a classifier that has good accuracy in the overall task of returning the maximum of two MNIST digits, but has low accuracy in classifying MNIST digits. The work in [40] showed that current NESY techniques are vulnerable to reasoning shortcuts. However, the work does not provide (class-specific) error bounds or any theoretical characterization of learning imbalances. The authors in [67] proposed necessary and sufficient conditions that ensure learnability of MI-PLL and provided error bounds for a state-of-the-art neurosymbolic loss under approximations [20]. Our theoretical analysis extends the analysis in [67] by providing (i) class-specific risk bounds (in contrast to [67], which only bounds $R(f)$) and (ii) stricter bounds for $R(f)$. In particular, as we show in Proposition 3.4, we can recover the bound from Lemma 1 in [67] by relaxing (2).

**Long-tailed learning**. The term *long-tailed learning* has been used to describe settings in which instances of some classes occur very frequently in the training set, with other classes being underrepresented. The problem has received considerable attention in supervised learning, with the proposed

techniques operating at training or testing time. Techniques in the former category typically work by reweighting the losses computed using the original training samples [6, 60, 59] or by over- or under-sampling during training [9, 4]. The techniques in the latter category work by modifying the classifiers' scores at testing time and using the modified scores for classification [25, 46], with LA being one of the most well-known techniques [42]. LA modifies the classifier's scores during testing time by subtracting the (unknown) gold ratios. In particular, the prediction of the classifier $f$ given input $x$ is given by $arg\max_{j\in[c]} f^j(x) - \ln(r_j)$. Our empirical analysis shows that CAROT is more effective than LA.

The most relevant to our work is the study in [46]. Unlike CAROT, the authors in [46] focus on PLL and use an optimal transport formulation [48] to adjust the scores of the classifier assuming that the marginal $\mathbf{r}$ is known. In contrast, CAROT relies on the assumption that $\widehat{\mathbf{r}}$ may be noisy, resorting to a robust optimal transport formulation [26] to improve the classification accuracy in these cases.

**PLL.** In PLL [10, 33, 14], each training sample is a tuple of the form $(x, \{l_1, \ldots, l_n\})$, where $x \in \mathcal{X}$ and $l_1, \ldots, l_n$ is a set of candidate, mutually exclusive labels for $x$ that includes the gold label of $x$. Since (1) each NESY training sample is represented as a tuple of the form $\mathbf{x}, \sigma^{-1}(s)$, see Section 2, where each element in $\sigma^{-1}(s)$ is a vector of candidate labels for the elements in $\mathbf{x}$, (2) the vectors in $\mathbf{x}, \sigma^{-1}(s)$ are mutually exclusive, and (3) $\sigma^{-1}(s)$ includes the gold labels $\mathbf{y}$ for the elements in $\mathbf{x}$, we can see that PLL reduces NESY(and MI-PLL) when restricting to input vectors of one label only.

The observation that certain classes are harder to learn than others dates back to the work of [10] in the context of PLL. We are the first to provide such results for NESY, also unveiling the relationship between $\sigma$ and class-specific risks.

**Long-tailed PLL.** A few recently proposed papers lie in the intersection of long-tailed learning and standard PLL, namely [32], RECORDS [18] and SOLAR [65], with the first one focusing on non-deep learning settings. RECORDS modifies the classifier's scores following the same idea with LA and uses the modified scores for training. However, it uses a momentum-updated prototype feature to estimate $\widehat{\mathbf{r}}$. RECORDS's design allows it to be used with any loss function and be trivially extended to support NESY. Our empirical analysis shows that RECORDS is less effective than CAROT, leading to lower classification accuracy when the same loss is adopted during training.

SOLAR shares some similarities with LP. In particular, given single-instance PLL samples of the form $\{(x_1, S_1), \ldots, (x_n, S_n)\}$, where each $S_\ell \subseteq \mathcal{Y}$ is the weak label of the $\ell$-th PLL sample[4], SOLAR finds pseudolabels $\mathbf{Q}$ by solving the following linear program:

$$\min_{\mathbf{Q}\in\Delta} \langle \mathbf{Q}, -\log(\mathbf{P})\rangle \tag{41}$$

$$\text{s.t.} \quad \Delta := \left\{ [q_{\ell,j}]_{n\times c} \mid \mathbf{Q}^\mathsf{T}\mathbf{1}_n = \widehat{\mathbf{r}},\ \mathbf{Q}\mathbf{1}_c = \mathbf{c},\ q_{\ell,j} = 0 \text{ if } j \notin S_\ell \right\} \subseteq [0,1]^{n\times c}$$

The program (41) shows that the information of each weak label $S_\ell$ is strictly encoded into $\Delta$. To directly extend (41) to NESY, we have two options:

- Use an $n \times c^M$ tensor $\mathbf{P}$ to store the scores of the classifier, where the cell $P[\ell, j_1, \ldots, j_c]$ stores the scores of the classifier for the label vector $(j_1, \ldots, j_c)$ associated with the $\ell$-th training NESY sample, for $1 \leq \ell \leq n$. However, that formulation would require an excessively large tensor, especially when $M$ becomes larger.

- Use separate tensors $\mathbf{P}_1, \ldots, \mathbf{P}_M$ to represent the model's scores of the $M$ instances, and set for each $1 \leq \ell \leq n$, the product $P_1[\ell, j_1] \times \cdots \times P_M[\ell, j_c]$ to be 0 if $(j_1, \ldots, j_c)$ does not belong to $\sigma^{-1}(s_\ell)$. However, that formulation would lead to a non-linear program.

Neither choice is scalable for NESY when $M$ is large[5]. Our work overcomes these issues by translating the information in the weak labels into linear constraints, leading to an LP formulation. Another difference between SOLAR and our work is that we developed Algorithm 1to estimate the ratios of the hidden labels, while SOLAR employs a window averaging technique to estimate $\mathbf{r}$ based on the model's scores [65]. Finally, although CAROT also uses a linear programming formulation with a Sinkhorn-style procedure, it differs from SOLAR in that it adjusts the classifier's scores at testing time rather than assigning pseudolabels at training time.

---

[4]In standard PLL, each weak label is a subset of classes from $\mathcal{Y}$.

[5]Yet another non-linear formulation is presented in Section D based on RSOT (see Section A).

**Listing 1** Theory for the Smallest Parent benchmark.

```
land_transportation :- automobile, truck
other_transportation :- airplane, ship
transportation :- land_transportation, other_transportation
home_land_animal :- cat, dog
wild_land_animal :- deer, horse
land_animal :- home_land_animal, wild_land_animal
other_animal :- bird, frog
animal :- land_animal, other_animal
entity :- transportation, animal
```

**Constrained learning.** NESY is closely related to constrained learning, in the sense that the predicted label vector $\mathbf{y}$ should adhere to the constraint $\sigma(\mathbf{y}) = s$. Training classifiers under constraints has been well studied in NLP [58, 49, 47, 43, 63, 68, 16, 41]. The work in [50] proposes a formulation for training under linear constraints; [54] proposes a Unified Expectation Maximization (UEM) framework that unifies several techniques, including CoDL [7] and Posterior Regularization [15]. The UEM framework was also adopted by [28] for NSL. Our LP formulation is orthogonal to UEM – it could be integrated with UEM, though.

The theoretical framework for constrained learning in [66] provides a generalization theory that suggests that encoding the constraints during both training and testing results in a better model compared to encoding the constraints only during testing. This theory could be extended to explain the advantages of LP-based techniques and to characterize the necessary conditions for CAROT to improve model performance.

**Other weakly supervised settings.** Another well-known weakly supervised learning setting is that of Multi-Instance Learning (MIL). In MIL, instances are not individually labelled but grouped into sets that contain at least one positive instance, or only negative instances, and the aim is to learn a *bag classifier* [52, 51]. In contrast, in NESY, instances are grouped into tuples, with each tuple of instances being associated with a set of mutually exclusive label vectors, and the aim is to learn an *instance classifier*.

# F  Further Experiments and Details

**Why using SL and Scallop.** SL [71, 37] has become the state-of-the-art approach to train deep classifiers in NSL settings. Training under SL requires computing a Boolean formula $\phi$ encoding all the possible label vectors in $\sigma^{-1}(s)$ for each NESY training sample $(\mathbf{x}, s)$ and then computing the weighted model counting [8] of $\phi$ given the softmax scores of $f$. SL has been effective in several tasks, including visual question answering [20], video-to-text alignment [30], and fine-tuning language models [29], and has nice theoretical properties [67, 40]. Due to its effectiveness, SL is now adopted by several NSL engines, such as DeepProbLog [37], DeepProbLog's successors [38], and Scallop [20, 30].

Our empirical analysis only uses Scallop, since it is the only engine that provides a scalable SL implementation that can support our scenarios when $M \geq 3$. The computation of $\sigma^{-1}(s)$ is generally required by NSL techniques [28, 37, 12, 72]. This computation can become a bottleneck when the space of candidate label vectors grows exponentially, as in our MAX-$M$, SUM-$M$, and HWF-$W$ scenarios. As also experimentally shown by [61, 67], the NESY techniques from [37, 38, 12, 28, 72] either time out after several hours while trying to compute $\sigma^{-1}(s)$, or lead to deep classifiers of much worse accuracy than Scallop. So, Scallop was the only engine that could support our experiments, balancing runtime with accuracy.

A further discussion about scalability issues in NESY can be found in Sections 3.2 and 6 at [67].

**Additional scenarios.** We additionally carried experiments with two other scenarios that have been widely used as NESY benchmarks, namely SUM-$M$ [36, 20] and HWF-$M$ [28, 30]. SUM-$M$ is similar to MAX-$M$, however, instead of taking the maximum, we take the sum of the gold labels. The HWF-$M$ scenario[6] was introduced in [27]. In this scenario, each training sample $((x_1, \ldots, x_M), s)$

---

[6]The benchmark is available at https://liqing.io/NGS/.

consists of a sequence $(x_1, \ldots, x_M)$ of digits in $\{0, \ldots, 9\}$ and mathematical operators in $\{+, -, *\}$, corresponding to a valid mathematical expression, where $s$ is the result of the mathematical expression. As in SUM-$M$, the goal is to train a classifier to recognize digits and mathematical operators. Notice that this benchmark is not i.i.d. since only specific types of input sequences are valid. The benchmark comes with a list of training samples. However, we created our own samples, to introduce imbalances in the distributions of the digits and operators.

**Computational infrastructure.** The experiments ran on an 64-bit Ubuntu 22.04.3 LTS machine with Intel(R) Xeon(R) Gold 6130 CPU @ 2.10GHz, 3.16TB hard disk and an NVIDIA GeForce RTX 2080 Ti GPU with 11264 MiB RAM. We used CUDA version 12.2.

**Software packages.** Our source code was implemented in Python 3.9. We used the following python libraries: `scallopy`[7], `highspy`[8], `or-tools`[9], `PySDD`[10], `PyTorch` and `PyTorch vision`. Finally, we used part of the code[11] available at [18] to implement RECORDS and part of the code[12] available at [65] to implement the sliding window approximation for marginal estimation.

**Classifiers.** For MAX-$M$ and SUM-$M$, we used the MNIST CNN also used in [20, 36]. For HWF-$M$, we used the CNN also used in [28, 30]. For Smallest Parent, we used the ResNet model also used in [65, 18].

**Data generation.** To create datasets for MAX-$M$, Smallest Parent, SUM-$M$, and HWF-$M$ we adopted the approach followed in previous work [12, 61, 67, 37, 20]. In particular, to create each training sample, we drew instances $x_1, \ldots, x_M$ independently by MNIST or CIFAR-10. Then, we applied the function $\sigma$ over the gold labels $y_1, \ldots, y_M$ to obtain the weak label $s$. To create samples for HWF-$M$, we followed similar steps to the above. However, to ensure that the input vectors of images represent a valid mathematical expression, we split the training instances into operators and digits, drawing instances of digits for odd $i$s and instances of operators for even $i$s, for $i \in [M]$. Before sample creation, the images in HWF were split into training and testing ones with ratio 70%/30%, as the benchmark does not offer such splits. As we state in Section 5, to simulate long-tail phenomena (denoted as **LT**), we vary the imbalance ratio $\rho$ of the distributions of the input instances as in [6, 65]: $\rho = 0$ means that the hidden label distribution is unmodified and balanced. In each scenario, the test data follows the same distribution as the hidden labels in the training NeSy data, e.g., when $\rho = 0$, the test data is balanced; otherwise, it is imbalanced under the same $\rho$.

**Further details.** For the Smallest Parent scenarios, we computed SL and (5) using the whole pre-image of each weak label. For the MAX-$M$ scenarios, we only consider the top-1 proof [67] both when running Scallop and in (5) as the space of pre-images is very large. For the Smallest Parent benchmark, we created the hierarchical relations shown in Listing 1 based on the classes from CIFAR-10.

**Tables and plots.** To assess the robustness of our techniques, we focus on scenarios with high imbalances, large number of input instances, and few NeSy training samples. Table 2 shows results for SUM-$M$, for $M \in \{5, 6, 7\}$, $\rho = \{50, 70\}$, and $m_\mathsf{P} = 2000$. Table 3 shows results for the same experiment, but $m_\mathsf{P} = 1000$. In Tables 3, LP(ALG1) refers to running LP using the gold ratios– Algorithm 1 cannot be applied, as the data is not i.i.d. in this scenario. Table 3 focuses on training time mitigation. RECORDS was not considered, as it led to substantially lower accuracy in the MAX-$M$ and Smallest Parent scenarios. Figure 8 shows the marginal estimates computed by Algorithm 1 for different scenarios. Last, Table 4 presents all the results for the MAX-$M$ scenarios. The tables follow the same notation with the ones in the main body of the paper.

**Conclusions.** The conclusions that we can draw from Tables 2, 3, and Figure 8 are very similar to the ones drawn in the main body of our paper. When LP is adopted jointly with the estimates obtained by Algorithm 1, we can see that the accuracy improvements are substantial on multiple occasions. For example, in SUM-6 with $\rho = 50$, the accuracy of classification increases from 67% under SL to 80% under LP(ALG1); in HWF-7 with $\rho = 15$, classification accuracy increases from 37% under SL to 41% under LP(ALG1). We argue that this is due to the low quality of the empirical estimates

---

[7] https://github.com/scallop-lang/scallop (MIT license).

[8] https://pypi.org/project/highspy/ (MIT license).

[9] https://developers.google.com/optimization/ (Apache-2.0 license).

[10] https://pypi.org/project/PySDD/ (Apache-2.0 license).

[11] https://github.com/MediaBrain-SJTU/RECORDS-LTPLL (MIT license).

[12] https://github.com/hbzju/SoLar.

Table 2: Experimental results for SUM-$M$ using $m_\text{P} = 2000$.

| Algorithms | **LT** $\rho = 50$ | | | **LT** $\rho = 70$ | | |
| --- | --- | --- | --- | --- | --- | --- |
| | $M = 5$ | $M = 6$ | $M = 7$ | $M = 5$ | $M = 6$ | $M = 7$ |
| SL | $82.28 \pm 15.87$ | $67.60 \pm 13.43$ | $88.42 \pm 15.66$ | $85.43 \pm 11.49$ | $85.60 \pm 12.36$ | $79.05 \pm 13.31$ |
| + LA | $81.74 \pm 16.27$ | $67.04 \pm 13.27$ | $78.33 \pm 15.61$ | $85.38 \pm 11.58$ | $85.47 \pm 12.49$ | $68.95 \pm 12.91$ |
| + CAROT | $82.21 \pm 15.94$ | $68.82 \pm 12.61$ | $79.54 \pm 14.46$ | $86.12 \pm 11.80$ | $85.47 \pm 12.37$ | $76.08 \pm 7.70$ |
| LP(ALG1) | $89.86 \pm 8.54$ | $80.10 \pm 18.45$ | $87.94 \pm 10.72$ | $91.64 \pm 7.62$ | $91.52 \pm 7.24$ | $63.79 \pm 12.97$ |
| + LA | $89.72 \pm 8.68$ | $79.43 \pm 19.15$ | $87.61 \pm 11.05$ | $91.66 \pm 7.60$ | $91.52 \pm 7.24$ | $63.70 \pm 12.87$ |
| + CAROT | $89.14 \pm 9.16$ | $78.85 \pm 19.55$ | $77.74 \pm 19.69$ | $91.29 \pm 7.86$ | $91.97 \pm 6.80$ | $67.06 \pm 9.78$ |

Table 3: Experimental results for HWF-$M$ using $m_\text{P} = 1000$.

| Algorithms | **LT** $\rho = 15$ | | |
| --- | --- | --- | --- |
| | $M = 3$ | $M = 5$ | $M = 7$ |
| SL | $94.01 \pm 0.49$ | $95.34 \pm 0.14$ | $48.23 \pm 6.91$ |
| LP(EMP) | $84.27 \pm 10.01$ | $84.86 \pm 10.80$ | $50.90 \pm 12.17$ |
| LP(GOLD) | $94.39 \pm 0.27$ | $95.72 \pm 0.34$ | $55.73 \pm 6.12$ |

of $\mathbf{r}$, a phenomenon that gets magnified due to the adopted approximations– recall that we run for SL and LP using the top-1 proofs, in order to make the computation tractable. The lower accuracy of LP(ALG1) for SUM-7 and $\rho = 70$ is attributed to the fact that the marginal estimates computed by Algorithm 1 diverge from the gold ones – see Figure 8. In fact, computing marginals for this scenario is particularly challenging due to the very large pre-image of $\sigma$ when $M = 7$, the high imbalance ratio ($\rho = 70$), and the small number of NESY samples ($m_\text{P} = 2000$). Table 3 also suggests that SOLAR's empirical ratio estimation technique may harm the accuracy of our LP-based formulation, supporting a claim that we also made in the main body of the paper, namely that *the computation of the marginals for training time mitigation is an important direction for future research.*

Figure 8 shows the robustness of Algorithm 1 in computing marginals. Figure 9 shows the hidden label ratios and the corresponding class-specific classification accuracies under the MAX-$M$ and the Smallest Parent scenarios for $\rho = 50$.

Table 4: Experimental results for MAX-$M$ using $m_P = 3000$.

| Algorithms | Original $\rho = 0$ | | | LT $\rho = 5$ | | | LT $\rho = 15$ | | | LT $\rho = 50$ | | |
|---|---|---|---|---|---|---|---|---|---|---|---|---|
| | $M=3$ | $M=4$ | $M=5$ | $M=3$ | $M=4$ | $M=5$ | $M=3$ | $M=4$ | $M=5$ | $M=3$ | $M=4$ | $M=5$ |
| SL | $84.15 \pm 11.92$ | $73.82 \pm 2.36$ | $59.88 \pm 5.58$ | $55.48 \pm 23.23$ | $66.24 \pm 1.22$ | $55.13 \pm 4.20$ | $71.25 \pm 4.48$ | $66.98 \pm 3.2$ | $55.06 \pm 5.21$ | $66.74 \pm 5.42$ | $67.71 \pm 11.58$ | $55.74 \pm 2.58$ |
| + LA | $84.17 \pm 11.95$ | $73.82 \pm 2.36$ | $59.88 \pm 5.58$ | $55.48 \pm 23.23$ | $65.63 \pm 1.75$ | $55.13 \pm 4.20$ | $70.80 \pm 4.52$ | $66.98 \pm 3.20$ | $54.53 \pm 5.74$ | $66.57 \pm 5.09$ | $61.10 \pm 3.95$ | $52.47 \pm 8.06$ |
| + CAROT | $84.57 \pm 11.50$ | $73.08 \pm 3.10$ | $60.26 \pm 5.20$ | $56.52 \pm 21.70$ | $66.70 \pm 0.76$ | $55.91 \pm 3.42$ | $74.95 \pm 3.45$ | $67.44 \pm 2.74$ | $55.80 \pm 4.47$ | $68.16 \pm 4.00$ | $68.25 \pm 6.14$ | $57.29 \pm 14.17$ |
| RECORDS | $85.56 \pm 7.25$ | $75.11 \pm 0.77$ | $59.43 \pm 6.61$ | $77.98 \pm 3.13$ | $65.85 \pm 0.62$ | $55.07 \pm 4.24$ | $55.47 \pm 20.45$ | $53.34 \pm 16.66$ | $52.40 \pm 7.95$ | $70.20 \pm 7.65$ | $66.05 \pm 13.90$ | $59.93 \pm 4.86$ |
| + LA | $87.63 \pm 5.11$ | $75.11 \pm 0.77$ | $59.28 \pm 6.76$ | $77.98 \pm 3.13$ | $65.43 \pm 0.87$ | $54.40 \pm 4.44$ | $54.90 \pm 20.16$ | $54.46 \pm 15.54$ | $51.25 \pm 9.09$ | $70.09 \pm 7.26$ | $65.78 \pm 14.18$ | $59.93 \pm 4.86$ |
| + CAROT | $90.97 \pm 2.03$ | $75.94 \pm 0.91$ | $60.45 \pm 7.78$ | $78.31 \pm 4.00$ | $67.57 \pm 1.74$ | $55.46 \pm 3.94$ | $54.32 \pm 21.85$ | $62.74 \pm 8.14$ | $55.85 \pm 4.61$ | $71.46 \pm 6.4$ | $71.25 \pm 8.70$ | $63.64 \pm 5.92$ |
| LP(EMP) | $94.97 \pm 1.32$ | $77.86 \pm 4.22$ | $55.27 \pm 11.27$ | $80.15 \pm 1.69$ | $70.73 \pm 1.85$ | $56.28 \pm 2.03$ | $75.83 \pm 5.26$ | $69.67 \pm 5.47$ | $59.25 \pm 7.27$ | $77.16 \pm 3.46$ | $70.06 \pm 10.73$ | $56.79 \pm 1.58$ |
| + LA | $94.69 \pm 1.60$ | $77.91 \pm 4.16$ | $55.34 \pm 11.19$ | $80.08 \pm 1.55$ | $70.54 \pm 1.82$ | $55.31 \pm 3.27$ | $75.77 \pm 5.32$ | $68.92 \pm 3.96$ | $58.49 \pm 5.74$ | $77.1 \pm 3.52$ | $69.76 \pm 10.31$ | $56.81 \pm 1.56$ |
| + CAROT | $95.07 \pm 1.20$ | $75.53 \pm 7.42$ | $53.07 \pm 12.99$ | $80.29 \pm 2.33$ | $70.88 \pm 2.22$ | $57.85 \pm 4.05$ | $76.38 \pm 4.72$ | $69.74 \pm 5.51$ | $59.56 \pm 8.14$ | $77.58 \pm 3.04$ | $70.11 \pm 10.34$ | $57.09 \pm 1.90$ |
| LP(ALG1) | $96.09 \pm 0.41$ | $78.34 \pm 4.80$ | $59.91 \pm 6.63$ | $78.56 \pm 1.52$ | $69.71 \pm 0.03$ | $57.61 \pm 3.09$ | $74.51 \pm 9.13$ | $69.14 \pm 1.82$ | $56.81 \pm 3.74$ | $72.23 \pm 11.49$ | $69.28 \pm 11.78$ | $63.67 \pm 7.04$ |
| + LA | $95.81 \pm 0.74$ | $78.97 \pm 4.09$ | $59.98 \pm 6.56$ | $78.48 \pm 1.53$ | $69.71 \pm 0.03$ | $57.47 \pm 3.09$ | $74.26 \pm 9.06$ | $68.73 \pm 2.23$ | $56.37 \pm 3.13$ | $72.23 \pm 11.49$ | $69.21 \pm 11.86$ | $63.67 \pm 7.04$ |
| + CAROT | $96.13 \pm 0.38$ | $80.78 \pm 2.36$ | $59.71 \pm 6.35$ | $78.93 \pm 1.85$ | $70.32 \pm 0.86$ | $57.62 \pm 3.08$ | $77.05 \pm 7.00$ | $69.19 \pm 1.81$ | $59.76 \pm 7.24$ | $74.82 \pm 10.18$ | $74.30 \pm 7.54$ | $64.39 \pm 6.43$ |

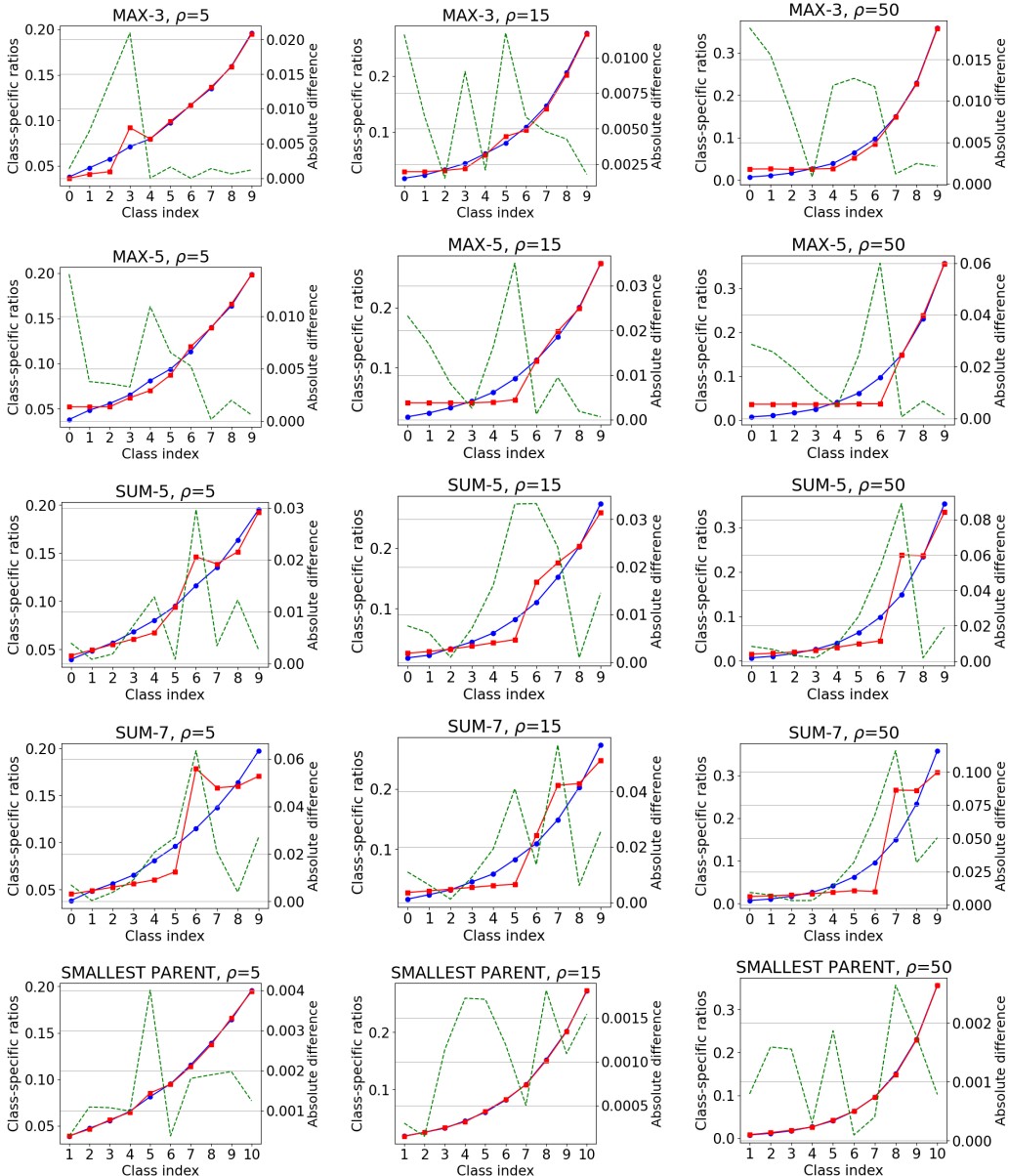

Figure 8: Accuracy of the marginal estimates computed by Algorithm 1 for different scenarios. Blue denotes the gold ratios, red the estimated ones, and green the absolute difference between the gold and estimated ratios.

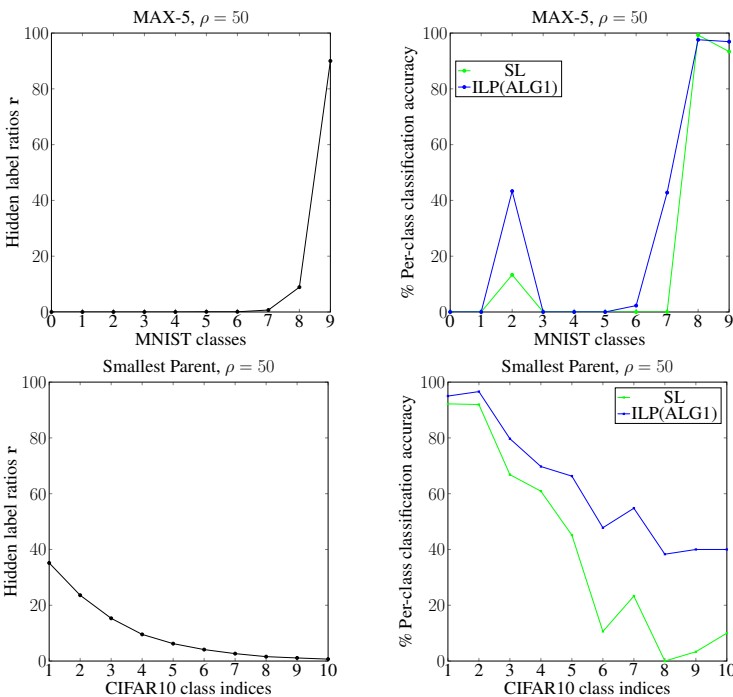

Figure 9: (Up left) hidden label ratios **r** for MAX-5 with $\rho = 50$. (Up right) Class-specific classification accuracies under SL and ILP(ALG1) for MAX-5 with $\rho = 50$. (Down left) hidden label ratios **r** for Smallest parent with $\rho = 50$. (Down right) Corresponding class-specific classification accuracies under SL and ILP(ALG1) for Smallest parent with $\rho = 50$.

Table 5: The notation in the preliminaries and the theoretical analysis.

| Supervised learning | |
|---|---|
| $1\{\cdot\}$ | Indicator function |
| $[n] := \{1, \ldots, n\}$ | Set notation |
| $\mathcal{X}, \mathcal{Y} = [c]$ | Input instance space and label space |
| $x, y$ | Elements from $\mathcal{X}$ and $\mathcal{Y}$ |
| $X, Y$ | Random variables over $\mathcal{X}$ and $\mathcal{Y}$ |
| $\mathcal{D}, \mathcal{D}_X, \mathcal{D}_Y$ | Joint distribution of $(X, Y)$ and marginals of $X$ and $Y$ |
| $r_j = \mathbb{P}(Y = j)$ | probability of occurrence (or ratio) of label $j \in \mathcal{Y}$ in $\mathcal{D}$ |
| $\mathcal{D}_Y := \mathbf{r} = (r_1, \ldots, r_c)$ | Marginal of $Y$ |
| $\Delta_c$ | Space of probability distributions over $\mathcal{Y}$ |
| $f : \mathcal{X} \to \Delta_c$ | Scoring function |
| $f^j(x)$ | Score of $f$ upon $x$ for class $j \in \mathcal{Y}$ |
| $[f] : \mathcal{X} \to \mathcal{Y}$ | Argmax classifier induced by $f$ |
| $\mathcal{F}, [\mathcal{F}]$ | Space of scoring functions and corresponding space of classifiers |
| $d_{[\mathcal{F}]}$ | Natarajan dimension of $[\mathcal{F}]$ |
| $L(y', y) := 1\{y' \neq y\}$ | Zero-one loss given $y, y' \in \mathcal{Y}$ |
| $R(f)$ | Zero-one risk of $f$ |
| $R_j(f) := P([f](x) \neq j \mid Y = j)$ | Risk of $f$ for the $j$-th class in $\mathcal{Y}$ |
| $D(\mathbf{A})$ | The diagonal matrix that shares the same diagonal with square matrix $\mathbf{A}$ |

| NESY | |
|---|---|
| $M > 0$ | Number of input instances per NESY sample |
| $\mathbf{x} = (x_1, \ldots, x_M), \mathbf{y} = (y_1, \ldots, y_M)$ | Vector of input instances and their (hidden) gold label |
| $\mathcal{S} = \{a_1, \ldots, a_{c_S}\}$ | Space of $c_S$ weak labels |
| $S$ | Random variable over $\mathcal{S}$ |
| $\sigma : \mathcal{Y}^M \to \mathcal{S}$ | Symbolic component (known to the learner) |
| $s = \sigma(\mathbf{y})$ | Weak label |
| $\sigma^{-1}(s)$ | Pre-image of $s$, i.e., set of all vectors $\mathbf{y} \in \mathcal{Y}^M$ s.t. $\sigma(\mathbf{y}) = s$ |
| $(\mathbf{x}, s)$ | NESY sample |
| $\mathcal{D}_\mathsf{P}$ | Distribution of NESY samples over $\mathcal{X}^M \times \mathcal{S}$ |
| $\mathcal{D}_{\mathsf{P}_S}$ | Marginal of $S$ |
| $\mathcal{T}_\mathsf{P}$ | Set of $m_\mathsf{P}$ NESY samples |
| $[f](\mathbf{x})$ | Short for $([f](x_1), \ldots, [f](x_M))$ |
| $L_\sigma(\mathbf{y}, s) := L(\sigma(\mathbf{y}), s)$ | Zero-one partial loss subject to $\sigma$ |
| $R_\mathsf{P}(f; \sigma) := E_{(X_1, \ldots, X_M, S) \sim \mathcal{D}_\mathsf{P}}[L_\sigma(([f](\mathbf{X})), S)]$ | Zero-one partial risk subject to $\sigma$ |
| $\widehat{R}_\mathsf{P}(f; \sigma, \mathcal{T}_\mathsf{P})$ | Empirical zero-one partial risk subject to $\sigma$ given set $\mathcal{T}_\mathsf{P}$ of NESY samples |

| Notation in Section 3 | |
|---|---|
| $\mathbf{1}_n, \mathbf{0}_n$ | All-one and all-zero vectors |
| $\mathbf{I}_n$ | Identity matrix of size $n \times n$ |
| $\mathbf{e}_j$ | $c$-dimensional one-hot vector, where the $j$-th element is one |
| $\mathbf{H}(f)$ | $c \times c$ matrix where the $(i, j)$ cell is the probability of $f$ classifying an instance with label $i \in \mathcal{Y}$ to $j \in \mathcal{Y}$. |
| $\mathbf{h}(f) := \mathrm{vec}(\mathbf{H}(f))$ | Vectorization of $\mathbf{H}(f)$ |
| $\mathbf{w}_j := \mathrm{vec}(\mathbf{W}_j)$ | Vectorization of matrix $\mathbf{W}_j := (\mathbf{1}_c - \mathbf{e}_j)\mathbf{e}_j^\mathsf{T}$, where $j \in \mathcal{Y}$ |
| $\mathbf{\Sigma}_{\sigma, \mathbf{r}}$ | Symmetric matrix in $R^{c^2 \times c^2}$ depending on $\sigma$ and $\mathbf{r}$ |
| $\Phi_{\sigma, j}(R_\mathsf{P}(f; \sigma))$ | Optimal solution to program (2) and upper bound to $R_j(f)$ |
| $\widetilde{R}_\mathsf{P}(f; \sigma, \mathcal{T}_\mathsf{P}, \delta)$ | Generalization bound of $R_\mathsf{P}(f; \sigma)$ for probability $1 - \delta$ |

Table 6: The notation used in our proposed algorithms.

**Notation in Section 4.1**

| | |
|---|---|
| $p_j := \mathbb{P}(S = a_j)$ | Probability of occurrence (or ratio) of $a_j \in \mathcal{S}$ in $\mathcal{D}_\mathsf{P}$ |
| $P_\sigma$ | System of polynomials $[p_j]_{j \in [c_S]}^\mathsf{T} = [\sum_{(y_1,\dots,y_M) \in \sigma^{-1}(a_j)} 1]_{j \in [c_S]}^\mathsf{T}$ |
| $\Psi_\sigma$ | Mapping of each $r_j \in \mathcal{Y}$ to its solution in $P_\sigma$, assuming $\mathbf{p}$ is known |
| $\widehat{\mathbf{r}}, \widehat{\mathbf{p}}$ | Estimates of $\mathbf{r}$ and $\mathbf{p}$ |
| $\bar{p}_j := \sum_{k=1}^{m_\mathsf{P}} \mathbb{1}\{s_k = a_j\}/m_\mathsf{P}$ | Estimate of $p_j$ given NESY dataset $\mathcal{T}_\mathsf{P}$ |

**Notation in Section 4.2**

| | |
|---|---|
| $n > 0$ | Size of each batch of NESY samples |
| $i$ | Index over $[M]$ |
| $j$ | Index over $[c]$ |
| $\ell$ | Index over $[n]$ |
| $(x_{\ell,1}, \dots, x_{\ell,M}, s_\ell)$ | $\ell$-th NESY training sample in the input batch |
| $R_\ell$ | Size of $\sigma^{-1}(s_\ell)$ |
| $t$ | Index over $[R_\ell]$ |
| $\mathbf{P}_i$ | Matrix in $[0,1]^{n \times c}$, where $P_i[\ell, j] = f^j(x_{\ell,i})$ |
| $\mathbf{Q}_i$ | Matrix in $[0,1]^{n \times c}$, where $Q_i[\ell, j]$ is the pseudo-label assigned with label $j \in \mathcal{Y}$ for instance $x_{\ell,i}$ |
| $q_{\ell,i,j}$ | A Boolean variable that is true if $x_{\ell,i}$ is assigned with label $j \in \mathcal{Y}$ and false otherwise |
| $\varphi_{\ell,t}$ | Conjunction over the $q_{\ell,i,j}$ Boolean variables that encodes the $t$-th label vector in $\sigma^{-1}(s_\ell)$ |
| $\Phi_\ell = \varphi_{\ell,1} \vee \cdots \vee \varphi_{\ell,R_\ell}$ | DNF formula encoding the label vectors in $\sigma^{-1}(s_\ell)$ |
| $\alpha_{\ell,t}$ | A fresh Boolean variable associated with each $\varphi_{\ell,t}$ by the Tseytin transformation |

**Notation in Section 4.3**

| | |
|---|---|
| $n > 0$ | Size of each batch of test input instances from $\mathcal{X}$ |
| $\mathbf{P}$ | Matrix in $R^{n \times c}$ of the $f$'s scores on the test instances of the input batch |
| $\mathbf{P}'$ | Matrix in $R^{n \times c}$ storing the CAROT's adjusted scores for $\mathbf{P}$ |
| $H(\mathbf{P}')$ | Entropy of $\mathbf{P}'$ |
| $\eta, \tau > 0$ | Parameters of robust semi-constrained optimal transport problem [26] |

