# OpenReview forum: "Imbalances in Neurosymbolic Learning: Characterization and Mitigating Strategies"
_NeurIPS.cc/2025/Conference — NeurIPS 2025 poster_

### Official Review · Reviewer_MSJR · 2025-06-20

**Clarity:** 3
**Significance:** 3
**Originality:** 2
**Rating:** 5
**Confidence:** 3

**Summary:**

This paper addresses the problem of mitigating class-specific error disparities in NeSy classification tasks. The analysis is sound and highlights how the concept-extractor function significantly influences these imbalances and propose mitigation strategies applicable at both training and test time. This paper support the approaches with experiments demonstrating improvements in classification accuracy.

**Questions:**

The topic of learning imbalances in NeSy is novel and relevant. However, I find it difficult to grasp the practical implications of mitigating these imbalances, especially in comparison to existing strategies for addressing Reasoning Shortcuts [1]. Could you provide concrete and realistic examples where your proposed strategy offers clear advantages? I believe that including a compelling example would significantly strengthen the paper.

Furthermore, how computationally demanding is the proposed linear programming approach during training? While the method appears to improve model accuracy, does it also lead to improvements in concept accuracy? The same question applies to the proposed test-time mitigation strategy.

Lastly, do you think the mitigation strategies you are proposing can be paired with those based on annotation acquisition like [2]?

------

**References**:

[1] Emanuele Marconato, Stefano Teso, Antonio Vergari, and Andrea Passerini. Not all neuro-symbolic concepts
are created equal: Analysis and mitigation of reasoning shortcuts. In NeurIPS, 2023
[2] Emanuele Marconato, Samuele Bortolotti, Emile van Krieken, Antonio Vergari, Andrea
Passerini, and Stefano Teso. BEARS Make Neuro-Symbolic Models Aware of their Reasoning
Shortcuts. In UAI, 2024

**Ethical Concerns:**

["NO or VERY MINOR ethics concerns only"]

**Final Justification:**

The rebuttal satisfied all my concerns

**Limitations:**

Yes

**Quality:**

3

**Strengths And Weaknesses:**

**Strengths**:

The paper presents a solid theoretical analysis that extends prior work by providing a bound on the risk associated with class-specific learning imbalances. This result is well-founded and offers a promising direction for improving the trustworthiness of NeSy models. Additionally, the authors propose a mitigation technique applicable during both training and test time.

------

**Weaknesses**:

The proposed mitigation strategies rely on linear programming and optimization procedures, which I believe may be challenging to scale in practice. Moreover, it is unclear how addressing class-specific imbalances interacts with existing methods aimed at solving reasoning shortcuts. Finally, while the theoretical contributions are valuable, the empirical evaluation is limited to relatively simple datasets such as MNIST and CIFAR.

---

> ### Author Rebuttal · Authors · 2025-07-31
>
> > **Comment 1**: The proposed mitigation strategies rely on linear programming and optimization procedures, which I believe may be challenging to scale in practice.
>
> First, our testing time mitigation strategy, CAROT, relies on a PTIME Sinkhorn algorithm, hence it is very efficient. Regarding a discussion on scalability and optimizations related to our LP formulation, please see our reply to Reviewer qjJg’ Comment 1 that discusses complexity aspects of our technique and relevant optimizations to reduce the runtime.
>
> Please also see our reply to Reviewer YWhF, Comment 2, where we provide runtimes for our LP formulation and the baseline SL. Furthermore, while our imbalance mitigation technique may mildly, in most cases, increase the training overhead as we empirically show, we would like to point the reviewers to similar issues that are encountered in other strategies for addressing the issue of ambiguous learning in NeSy and, in particular, to the mitigation strategies proposed in [1] and [2].
>
> Starting from [2], the proposed mitigation strategy involves training an ensemble of instance classifiers, which, as also acknowledged by the authors, in Section 3.6., increases the training time proportionally to the number of models in the ensemble. Instead, our formulation involves training a single classifier each time (with slightly increased runtime over the SOTA training loss). Second, the objective-based mitigation strategy in Section 5.3 from [1] involves training using an autoencoder which can be computationally expensive. As the authors state at the end of Section 5.3. “However, minimizing the reconstruction can be non-trivial in practice, especially for complex inputs”.
>
> Overall, we agree with the reviewer that our mitigation strategy may increase the training overhead, however, this may be inevitable to improve the model accuracy, as it is the case in relevant research as well.
>
> > **Comment 2**: Finally, while the theoretical contributions are valuable, the empirical evaluation is limited to relatively simple datasets such as MNIST and CIFAR.
>
> The MAX, SUM, HWF, and other benchmarks have been used by the most influential papers in the relevant community, e.g., [32,18,25,36]. Unlike them, however, we considered larger values for M (up to 7, see Table 3).
>
> **Results on VQAR.** We additionally conducted experiments on VQAR [18], a VQA benchmark based on GQA. VQAR exhibits two types of imbalances. First, the base classes are highly imbalanced. For example, the most frequent “name” class is “window” with 35.9K training instances, the 20th most frequent “name” class is “table” with 10K instances, while the 30th most frequent class is “nose” with 7.6K instances. Second, the type hierarchies, practically $\sigma$, further exacerbate those imbalances, e.g., there are five different warcraft types, but 75 different food types. As the training samples in VQAR are not i.i.d., we empirically approximated $\mathbf{r}$ using the moving window heuristic from [59]. In our experiments, we considered queries of the form “Q(O) :- name(O, superclass)”, where superclass is a superclass of a base class, e.g., fruit is a superclass for orange. The benchmark comes with 500 classes. We considered the whole $\sigma^{-1}$ to form the linear program. We ran experiments focusing on the 30 most frequent ones.
>
> For the baseline SL [65], the mean balanced accuracy over 3,000 training samples was 46.84\% for SL. For ILP, the mean balanced accuracy increased to 53.06\%. The three classes with biggest improvements were door, fence, and nose where the accuracies improved from 22.76% to 29.68%, from 19.32\% to 41.03\%, and from 32.83\% to 43.17\%.
>
> > **Comment 3**: The topic of learning imbalances in NeSy is novel and relevant. However, I find it difficult to grasp the practical implications of mitigating these imbalances, especially in comparison to existing strategies for addressing Reasoning Shortcuts [1].
>
> First, it is worth noting that reasoning shortcuts are essentially all the classifiers that minimize the weakly supervised loss, e.g., the semantic loss, but map the input instances to wrong concepts. Notice that the term “concept” in reasoning shortcuts corresponds to the term “instance label” in our notation.
>
> Second, many strategies for mitigating reasoning shortcuts require *additional* assumptions:
>
> 1. Knowledge-based Mitigation. Involves editing the symbolic component directly by, e.g., eliciting additional constraints from a domain expert.
> 2. Multi-Task Learning (MTL). Involves training a classifier using samples corresponding to different symbolic components, e.g., mixing M-MAX with M-SUM training samples.
> 3. Data-based Mitigation. Involves providing supervised samples with the weakly-supervised ones.
> 4. Architecture-based Mitigation. Involves using a more complex neural architecture.
> 5. Objective-based Mitigation. Uses an autoencoder along with the neural model.
>
> Instead, the focus of our work is to mitigate imbalances without making additional assumptions on the supervision. Nevertheless, we do not disagree that integrating these strategies with our imbalance mitigation techniques would potentially improve the classification accuracy. It’s like combining long-tailed learning techniques from standard deep learning with representation learning approaches, such as contrastive learning. They are proposed to tackle different aspects. Yet, their combination could be very effective. We will add this discussion in the revised paper.
>
> As in standard ML [4, 54, 53, 7, 2, 22, 42, 38], the practical implication of mitigating imbalances in NeSY is to force the classifier to predict minority classes at testing time or give more mass to minority classes at training time. This intuition is discussed in lines 183–192 of our submission.
>
> > **Comment 4**: Could you provide concrete and realistic examples where your proposed strategy offers clear advantages? I believe that including a compelling example would significantly strengthen the paper.
>
> About using a compelling example. We follow the same approach with the reasoning shortcuts paper, where they exemplify the notion of reasoning shortcuts via the MNIST-Addition (see their Example 1 and 2), and exemplify the concept of learning imbalances via the MNIST-MAX example.
>
> Concretely:
> - Figure 1 shows the accuracy of an MNIST classifier trained with MNIST-MAX training samples using the SL, suggesting the phenomenon of learning imbalances in NeSy.
> - Example 3.2 and the left part of Figure 2 show that our class-specific error bounds align with the observations in Figure 1.
> - The upper part of Figure 8 in the appendix compares the class-specific accuracies of a classifier trained using MNIST-MAX data for $M=5$ and $\rho=50$ under the SL (green curve) and our ILP technique (blue curve). This scenario is very challenging as the frequencies of the hidden labels are highly imbalanced: 0.0000e+00, 0.0000e+00, 6.67E-05, 0.0000e+00, 0.0000e+00, 6.0000e-04, 6.6667e-04, 7.3333e-03, 8.9667e-02, 9.0167e-01. In this scenario, the baseline model (row SL in Table 1, for $M=5$ and $\rho=50$) has the following class-specific classification accuracies:
> 0\%, 0\%, 13.33\%, 0\%, 0\%, 0\%, 0\%, 0\%, 99.28\%, 93.35\%.
> Our training-time mitigation technique (row ILP(ALG1) in Table 1, for $M=5$ and $\rho=50$) leads to the following class-specific classification accuracies:
> 0\%, 0\%, 43,33\%, 0\%, 0\%, 0\%, 2.29\%, 42.80\%, 97.61\%, 96.91\%,
> substantially improving the classification accuracy for the MNIST digits
> 2, 6, and 7 -- especially for 6 and 7, the baseline model has accuracy zero.
>
> As we wrote in reply to Comment 3, the practical implication of mitigating imbalances in NeSY is to force the classifier to predict minority classes during testing or give more mass to minority classes during training.
>
> As a realistic example, please see our earlier replies to your Comment 2, where we included additional experiment results for VQAR.
>
> > **Comment 5**: Furthermore, how computationally demanding is the proposed linear programming approach during training?
>
> Please see our response to Comment 1.
>
> > **Comment 6**: While the method appears to improve model accuracy, does it also lead to improvements in concept accuracy? The same question applies to the proposed test-time mitigation strategy.
>
> In our work, the model accuracy is evaluated based on class-specific accuracies, i.e., how well the classifier predicts the class of each input. In particular, the class-specific accuracies are balanced to ensure a fair comparison, see lines 306–308. In addition, in our setting, the term “concept” in the reasoning shortcuts terminology coincides with the term “instance label” in our terminology. Hence, by improving the model accuracy, we do improve the concept accuracy.
>
> > **Comment 7**: Lastly, do you think the mitigation strategies you are proposing can be paired with those based on annotation acquisition like [2]?
>
> This is a great point. We believe the reviewer refers to Section 3.5 in [2], which presents an active learning strategy for querying the most informative gold annotations: "requesting supervision only for specific concepts G_j by maximizing H(p(Cj |xi)) for i and j." Yes, it is possible to combine this active learning strategy with our training time mitigation technique. In particular, we could encourage the acquisition of labels for concepts that maximize the entropy and, at the same time, appear with smaller ratios. The latter can be achieved, for example, by assigning a higher weight to concepts with smaller (estimated) ratios in the entropy computation. We will add the relevant discussion in the revision of our paper.

---

> > ### Comment · Reviewer_MSJR · 2025-08-04
> >
> > Thank you for the detailed response and for the good work.
> >
> > I appreciate the clarifications and the effort you have put into addressing my concerns.
> >
> > I have accordingly increased my score by one point.

---

### Official Review · Reviewer_YWhF · 2025-06-26

**Clarity:** 2
**Significance:** 2
**Originality:** 3
**Rating:** 4
**Confidence:** 4

**Summary:**

The paper focuses on a neglected core issue in neurosymbolic learning (NSL)—learning imbalance, which refers to systematic differences in classification errors across different classes. Through theoretical analysis and algorithmic design, it reveals the independent influence of symbolic components on class imbalance in NSL. The authors propose two algorithms (LP and CAROT) to mitigate imbalance during training and testing phases, and validate their effectiveness through experiments. These algorithms achieve significant performance improvements over existing baseline methods.

**Questions:**

How is the NP-hardness of LP mitigated on large-scale datasets? Are approximation algorithms or parallelization considered?

**Ethical Concerns:**

["NO or VERY MINOR ethics concerns only"]

**Final Justification:**

We have read the reply to Comment 1 and are willing to increase the score.

**Limitations:**

The computational complexity of nonlinear programming can be further discussed, particularly regarding its scalability in high-dimensional scenarios.

**Quality:**

2

**Strengths And Weaknesses:**

Strengths

The paper provide a class-specific error bound in the context of NESY and show the effect of $\sigma$ on learning imbalances.

The authors propose two algorithms to mitigate imbalances during training and testing time.

Weaknesses

1.The dependence of CAROT on the accuracy of $r$ and the dependence of the LP method on batch size and $\sigma$ should be supplemented in the Discussion or Conclusion section with an analysis of these limitations.

2. The LP-based method requires solving a linear program for each training batch, which may incur substantial computational overhead, especially for large batch sizes. It is recommended to report the runtime and resource consumption in the main text and to provide a discussion on the scalability of the approach.

3. The main text (e.g., Section 2) references "Tables 6 and 7," but these tables are absent in the main manuscript. While they may be included in the appendix, this is not explicitly stated. Section 4 introduces numerous mathematical symbols, yet their explanations are deferred to Table 7, creating a disconnect that hampers readability.

4. The structure is densely packed, and transitions between concepts are abrupt. In particular, the shift from Section 3 to Section 4 is rather sudden. It is recommended to include a brief transitional paragraph to improve the flow.

---

> ### Author Rebuttal · Authors · 2025-07-31
>
> > **Comment 1**: The dependence of CAROT on the accuracy of r and the dependence of the LP method on batch size and \sigma should be supplemented in the Discussion or Conclusion section with an analysis of these limitations.
>
> - About the dependence of CAROT on $\mathbf{r}$, we do have such an analysis in Figure 4. The relevant discussion is in lines 348–353.
> - About the dependence of LP on $\sigma$, we want to stress that, due to exponential explosion in the number of pre-images in $\sigma^{-1}$ as $M$ increases (e.g., in the MAX-5 scenario, there are 5 × 94 candidate label vectors when the weak label is 9), we ran all experiments for M-MAX, M-SUM, and HWF both for the baseline SL using the top-$k$ pre-images, as we discuss in the paragraph starting in line 1201 in the appendix. The blow-up of enumerating $\sigma^{-1}$ is independent of SL and LP. Notice also that by increasing the top-$k$, the experiments would quickly become infeasible to run both for SL and LP, due to the very large cost for exploring the pre-images. In the Smallest Parent benchmark, we considered the whole $\sigma^{-1}$, as this exponential blowup was not occurring. To address this comment, we additionally conducted experiments on a new VQA benchmark based on GQA, VQAR [18], to assess LP's scalability on more realistic $\sigma$s. In this benchmark, like in Smallest Parent, we also considered the whole $\sigma^{-1}$. Detailed results on this benchmark are given in our replies to Reviewer MSJR’s Comment 2.
> - Regarding the dependence of our LP technique to the batch size and relevant measurements, we provide details to the Comment 2 just below.
>
> > **Comment 2**: The LP-based method requires solving a linear program for each training batch, which may incur substantial computational overhead, especially for large batch sizes. It is recommended to report the runtime and resource consumption in the main text. provide a discussion on the scalability of the approach.
>
> Below we provide results for all the additional experiments and measurements requested by the reviewer. A quick summary is as follows:
> 1. The baseline training approach and our LP technique are on par in terms of runtime and max memory consumption in most cases.
> 2. Our experiments in our original submission ran with a small batch size (64). By increasing it, the balanced accuracy increases.
>
> We start by showing results on the runtime for SL and ILP for different batch sizes. The results in our submission were computed based on batch size = 64. As discussed in Section 5, we used Scallop’s implementation [18] of the SL and or-tools (as mentioned in line 1182) to solve the LP. Further details about our implementation are in Appendix F.
>
> **Table A.** Mean runtime per epoch in minutes for MAX-$M$ (2,000 training samples). We form the LP based on the training samples in the current batch.
>
> | Technique |   M=3 |   M=4 |
> |-------------|--------------:|--------------:|
> | SL(batch size = 64)    [65]     |    7.14  |    7.23  |
> | SL(batch size = 128)   [65]      |    5.00  |    5.40  |
> | SL(batch size = 256)  [65]       |   2.26  |   2.12  |
> | SL(batch size = 512)    [65]     |   1.42  |   1.32  |
> | LP(batch size = 64)         |    7.16  |    8.21  |
> | LP(batch size = 128)           |    5.16  |    5.29  |
> | LP(batch size = 256)           |   3.12  |   3.28  |
> | LP(batch size = 512)           |    4.01  |   5.42  |
>
> **Max memory consumption.** The max memory consumption for the hardest case (MAX-4 for batch size = 512 and 2000 training samples) is:
> - SL [65]: 1.74 GB
> - LP: 1.80 GB
>
> **Table B.** Mean runtime per epoch in minutes for SUM-$M$ (1,000 training samples). We form the LP based on the training samples in the current batch.
>
> | Technique |   M=5 |   M=6 |
> |-------------|--------------:|--------------:|
> | SL(batch size = 64)    [65]     |    5.59  |    5.48  |
> | SL(batch size = 256)  [65]       |   0.43  |   0.57  |
> | SL(batch size = 512)    [65]     |   0.44  |   0.55  |
> | LP(batch size = 64)         |    6.06  |    6.28  |
> | LP(batch size = 256)           |   3.11  |  4.07  |
> | LP(batch size = 512)           |    6.00  |  7.40  |
>
> **Max memory consumption.** The max memory consumption for the hardest case (SUM-6 for batch size = 512 and 1000 training samples) is:
> -	SL [65]: 823 MB
> -	LP: 1.07 GB
>
> Below, we show the balanced mean accuracy for LP(ALG1) for different batch sizes. The results that we show in our submission, were computed based on batch size = 64.
>
> **Table C.** Balanced accuracy for MAX-$M$, $\rho$=50 (3,000 training samples).
>
> | Technique |   M=3 |   M=4 |
> |-------------|--------------:|--------------:|
> | LP(batch size = 64)         |    72.23 ± 11.49  |    69.28 ± 11.78  |
> | LP(batch size = 128)           |    72.98 ± 09.32  |    70.83 ± 08.45  |
> | LP(batch size = 256)           |    73.35 ± 08.27  |   71.03 ± 05.01  |
>
> **Table D.** Balanced accuracy for Smallest parent, $\rho$=50 (3,000 training samples).
>
> | Technique |   M=2 |
> |-------------|--------------:|
> | LP(batch size = 64)         |    83.44 ± 0.48  |
> | LP(batch size = 128)           |    84.41 ± 0.69  |
> | LP(batch size = 256)           |    85.87 ± 0.52  |
>
> > **Comment 3**: The main text (e.g., Section 2) references "Tables 6 and 7," but these tables are absent in the main manuscript. While they may be included in the appendix, this is not explicitly stated. Section 4 introduces numerous mathematical symbols, yet their explanations are deferred to Table 7, creating a disconnect that hampers readability.
>
> Indeed, Tables 6 and 7 are in the appendix. The corresponding symbols are all defined in the main body; however, to enhance readability, we summarized them in Tables 6 and 7. Unfortunately, as these tables are large, it is difficult to move them to the main body.
>
> > **Comment 4**: The structure is densely packed, and transitions between concepts are abrupt. In particular, the shift from Section 3 to Section 4 is rather sudden. It is recommended to include a brief transitional paragraph to improve the flow.
>
> The paragraph in lines 183–192 explains the transition between Section 3 and Section 4. Perhaps the reviewer means something else?

---

### Official Review · Reviewer_mHmX · 2025-06-30

**Clarity:** 3
**Significance:** 3
**Originality:** 3
**Rating:** 5
**Confidence:** 4

**Summary:**

This paper investigates learning imbalances in neurosymbolic learning (NSL), specifically focusing on class-specific risks that emerge when neural classifiers are trained with hidden gold labels revealed only through a symbolic component. Unlike previous research, this work highlights how the symbolic component itself can cause imbalances, not just data imbalances. The authors propose a method to estimate hidden gold label marginals and introduce algorithms to mitigate these imbalances during both training and testing, achieving up to 14% performance improvement over existing baselines.

**Questions:**

I notice that in the experiment part, you use only MAX-M to validate the effectiveness of your algorithm. Are there any other choices of $\sigma$ in practical scenarios? Have you considered the approximation of $\hat{\boldsymbol{r}}$ and effectiveness of CAROT in different choices of $\sigma$?

**Ethical Concerns:**

["NO or VERY MINOR ethics concerns only"]

**Final Justification:**

The authors addresses all of my concerns. This paper is overall in good quality as I have stated in "strengths" part of my review. I decide to keep my rating at 5.

**Limitations:**

Yes, the authors have adequately addressed the limitations and potential negative societal impact of their work.

**Quality:**

4

**Strengths And Weaknesses:**

## Strengths

1. The authors continuously use simplified and intuitive examples to illustrate the important comcepts and strategies, making the paper easy to read.
2. The paper introduces and addresses the unexplored problem of learning imbalances in Neurosymbolic Learning (NESY), which is novel and applicable.
3. This paper has strong theoretical foundations, offering stricter bounds and makes fewer assumptions compared to existing work.
4. Empirical experiments validates the effectiveness of the proposed CAROT algorithm.

## Weaknesses

1. Consider Case 1 in Example 3.2. The solution of (2), though tighter than exsiting bounds, are still worst-case upper bounds. It might not be appropriate to claim that class 0 is the hardest to learn by comparing the worst-case solutions of classification risks.
2. A minor issue: missing noun in line 8: "Our theoretical (?) reveals a unique phenomenon".

---

> ### Author Rebuttal · Authors · 2025-07-31
>
> > **Comment 1**: Consider Case 1 in Example 3.2. The solution of (2), though tighter than exsiting bounds, are still worst-case upper bounds. It might not be appropriate to claim that class 0 is the hardest to learn by comparing the worst-case solutions of classification risks.
>
> Thanks for the question. While worst-case upper bounds does not imply that class 0 would necessarily have the highest label-specific risks in **all** the training scenarios – since it depends on the model/data/initialization – however, the worst-case analysis can serve as a conservative, distribution-free way to characterize the learning difficulty, just like in mathematical analysis, one may use supreme norm to characterize the distance between two functions, when the pointwise comparison is intractable. Such worst-case comparisons are also standard in learning theory. For example, people use VC dimension and Rademacher complexity to understand relative model complexity/task difficulty when exact risks are intractable.
>
> On the other hand, our bound is tight in the sense that it can be achieved by constructing a classifier whose label-specific errors are equal to the maximizer of the program in Eq. (2). Therefore, we believe our theory serves as a meaningful description of the relationship between the partial risk and class-specific risk.
>
> Nonetheless, we can add a discussion on this point in the revised version to avoid misinterpretation.
>
> > **Comment 2**: A minor issue: missing noun in line 8: "Our theoretical (?) reveals a unique phenomenon".
>
> It should be “analysis”. Thanks for pointing it out.
>
> > **Comment 3** I notice that in the experiment part, you use only MAX-M to validate the effectiveness of your algorithm.
>
> We do consider other benchmarks, such as Smallest Parent (lower part of Table 1) and SUM-M (Table 2 in the appendix). We have empirically shown that CAROT can considerably improve the model’s accuracy for the corresponding $\sigma$s.
>
> > **Comment 4**: Are there any other choices of \sigma in practical scenarios?
>
> Regarding choices of $\sigma$ in practical scenarios, in the literature, people also considered other transitions such as sorting of numbers [32] and transitions induced by video to text alignment. We described several applications of our learning setting in lines 28 – 31. To address this comment, we additionally conducted experiments on a new VQA benchmark based on GQA, VQAR [18]. Detailed results on this benchmark are given in our replies to Reviewer MSJR’s Comment 2.
>
> With other choices of $\sigma$, our Algorithm 1 for estimating $\hat{\mathbf{r}}$ can be applied without modification as long as the i.i.d. assumption for the instances holds, and we believe that CAROT would also improve the model’s accuracy in such tasks if the transition itself introduces imbalance or the data is imbalanced.
>
> In more complex settings where the i.i.d. assumption does not hold, one could still apply [1] to approximate $\mathbf{r}$. However, this estimator can be biased, in general, depending on the strength/structure of the sample correlation. It will be an interesting future research direction to develop a more effective version of Algorithm 1 for cases where i.i.d. assumption does not hold. Notice that in the experiments we conducted on VQAR, we approximated $\mathbf{r}$ by moving window heuristic [59]. Despite the approximations, our technique offered good accuracy improvements, especially for the minority classes, e.g., for the baseline SL [65], the mean balanced accuracy over 3,000 training samples was 46.84\% for SL. For ILP, the mean balanced accuracy increased to 53.06\%. Further details are given in our replies to Reviewer MSJR’s Comment 2.
>
> > **Comment 5**: Have you considered the approximation of \hat{r} and effectiveness of CAROT in different choices of \sigma?
>
> Regarding the effectiveness of CAROT under different choices of $\sigma$, yes, CAROT has been applied in the MAX-M (upper part of Table 1), Smallest part (lower part of Table 1), and SUM-M (Table 2) scenarios.
>
> Regarding the methods we have tried to approximate $\hat{r}$, please see our response above for Comment 4. Regarding the effectiveness of CAROT under different approximations of $\mathbf{r}$, we analyze the sensitivity of CAROT subject to the quality of the input \hat{\mathbf{r}}. The results are shown in Figure 4 and discussed in lines 348 –353. We observed that CAROT’s effectiveness drops as the estimated marginal diverges more from $\mathbf{r}$.

---

> > ### Comment · Reviewer_mHmX · 2025-08-01
> >
> > Thank you for your response! I have read your rebuttal to all reviewers and I decide to keep my score.

---

### Official Review · Reviewer_qjJg · 2025-07-03

**Clarity:** 2
**Significance:** 2
**Originality:** 2
**Rating:** 4
**Confidence:** 3

**Summary:**

This paper addresses the challenge of mitigating class imbalance in neurosymbolic learning (NESY) systems by proposing CAROT, a distributionally robust optimization framework that adjusts for class-specific risk at both training and test time. The authors formalize class imbalance as a risk reweighting problem using a linear program and propose a batch-level approximation algorithm (Algorithm 1) to compute instance weights dynamically. The framework is evaluated across synthetic and real-world benchmarks (MNIST, CIFAR-10), and claims to improve balanced accuracy in low-data and long-tail settings. The theoretical contributions include bounding excess risk in class-imbalanced settings and analyzing generalization under distribution shift.

**Questions:**

1. Can the authors relax the i.i.d. assumption to allow for structural dependencies among instances (e.g., compositional overlap in symbolic trees)? How might this affect the guarantees in Propositions 3.1–3.4?
2. What is the sensitivity of Algorithm 1 to initializations, learning rate choices, or softmax reparameterization temperature? Were convergence pathologies (e.g., oscillations, mode collapse) observed?
3. Given that CIFAR-10 and MNIST are not symbolic reasoning datasets, what modifications would be needed to deploy CAROT on real NESY benchmarks (e.g., CLEVRER, NLVR, or hybrid QA tasks)? Have the authors considered adapting the framework to hierarchical or graph-structured label spaces?

**Ethical Concerns:**

["NO or VERY MINOR ethics concerns only"]

**Final Justification:**

As expressed below, I feel the authors have done a great job at addressing my concerns and I will be updating their rating to Borderline Accept.

**Limitations:**

The authors acknowledge several limitations (computational overhead, symbolic domain coverage). However, the assumptions regarding instance independence, class-only misclassification risk, and the representativeness of synthetic benchmarks warrant deeper discussion. In particular, the societal risks of deploying fairness-motivated reweighting without proper calibration or robustness checks in high-stakes symbolic systems such as legal or medical rule-based AI deserve consideration.

**Paper Formatting Concerns:**

1. Figures 3 and 4 contain axis labels that are difficult to read at standard print size. Consider enlarging font or adjusting resolution.
2. Include solver and hardware specs in the appendix to ensure reproducibility of LP-based algorithms.

**Quality:**

2

**Strengths And Weaknesses:**

Let's begin with Strengths:
The underlying issue of how NESY systems overfit to dominant symbolic patterns due to class imbalance in supervisory signals is an inadequately explored issue and I am glad the authors have done so in this paper. CAROT appears to be a thoughtful adaptation of DRO principles to symbolic learning. Of note, CAROT consistently improves balanced accuracy across synthetic and real-world datasets. In particular, it appears more robust than prior approaches like RECORDS and LA in long-tail settings. From a theoretical perspective,the generalization bounds in Section 3 provide formal support for the class-level risk minimization approach and the decomposability of the loss into class partitions seems well-leveraged.

Now, let's get into the Weaknesses:
1. Methodological Limitations
- The LP in Equation (5) can have exponential constraint growth (via σ⁻¹ enumeration). No concrete approximation strategies are proposed, which limits scalability. Possible alternatives (e.g., sampling-based surrogates, hierarchical grouping of symbolic types, streaming approximations) are not discussed.
- The softmax reparameterization in Algorithm 1 may cause optimization instability, particularly in skewed or underdetermined settings. No ablation is provided for this choice.
- CAROT assumes known symbolic partitions, which is restrictive. Real-world NESY systems often operate with evolving or noisy taxonomies. This limits generalizability.

2. Theoretical Gaps
- The assumption that misclassification depends only on class is strong. Symbolic tasks typically involve multi-factor dependencies (e.g., depth of composition, operator type). This abstraction may misrepresent error dynamics.
- The paper assumes i.i.d. symbolic structures, yet most NESY settings include structured dependencies (e.g., scene graphs, logic programs). It would strengthen the paper to either relax this assumption or justify it empirically.
- The tightness and operational utility of the theoretical bounds (Propositions 3.1–3.4) is unclear. Do they guide parameter selection or batch size decisions? No empirical validation is provided.

3. Experimental Limitations
- Evaluation is done on overly synthetic symbolic tasks (MAX, SUM, etc.) that may not reflect real-world NESY generalization challenges. The authors may benefit from considering evaluation on more complex symbolic reasoning datasets like CLEVRER, ProofWriter, or GQA.
- Only 3 seeds are used, which is insufficient given the observed variance across some datasets (e.g., CIFAR-10). For high-variance metrics like balanced accuracy, ≥10 seeds would improve statistical reliability.
- Baseline selection could be expanded. While RECORDS and LA are relevant, stronger fairness-focused baselines like IRM (“Invariant Risk Minimization”, Arjovsky et al., 2019) or Group DRO (“An Investigation of Why Overparameterization Exacerbates Spurious Correlations”, Sagawa et al., 2020) are notably absent. Even basic class-reweighting schemes are omitted.
- No computational benchmarks are provided. Training time overhead, LP solver configuration, and memory usage are critical in assessing CAROT’s deployability.

---

> ### Author Rebuttal · Authors · 2025-07-30
>
> > **Comment 1**: The LP in Eqn (5) can have exp constraint growth (via σ⁻¹ enumeration). No concrete approximation strategies proposed,... Possible alternatives (e.g., sampling-based surrogates, hierarchical grouping of symbolic types, streaming approximations) not discussed
>
> 1. LP does NOT grow exponentially in σ⁻¹. It has a LINEAR growth, as we employ the Tseytin transformation to translate from the pre-image into the ILP (clarified in line 246). The transformed formula's size grows linearly to the input's. Details of this transformation is in Appendix D
> 2. The complexity of enumerating σ⁻¹ is inherent to all relevant NeSy frameworks [32,18,65,25], as they all rely on the pre-image to compute a loss. This is discussed in Section 3.2 of [61]
> 3. In our experiments, we consider top-$k$ pre-images to address the crux of exhaustively enumerating the preimage. Other techniques, such as [25], rely on sampling
> 4. As discussed in line 383, the complexity of applying SOTA NeSy loss, SL [65], is \#P-complete [6]. In contrast, the complexity of solving our ILP is NP-hard
> 5. There are multiple techniques to solve ILP efficiently, e.g., “Local Branching Relaxation Heuristics for ILP” (Huang et al., 2023)
> 6. We could treat the program in Eqn 5 as a differentiable layer by applying optimization techniques as in “Differentiable Convex Optimization Layers” (Agrawal et al., 2019)
> 7. Eqn 5 is a LP when allowing the variables to take values in [0,1]. Then, we can employ the simplex algorithm which runs  quite efficiently in practice, see “Smoothed analysis of algorithms Why the simplex algorithm usually takes poly time” (Spielman et al., 2004)
>
> Since the literature in points 5-7 is orthogonal to our contribution (mitigating NeSy imbalances), we decided not to touch upon this topic. However, we'll include this discussion in the revised version.
>
> Finally, we are unsure what it means by “hierarchical grouping of symbolic types”. Could it be possible to clarify?
>
> > **Comment 2** on stability of Alg1 with softmax
>
> Regarding the sensitivity of Alg1 w/ softmax reparameterization, we did additional sensitivity analysis for M-MAX with M=3. We consider a range of LR in {0.1, 0.01, 0.001, 0.0001} and temperatures in {0.1, 0.5, 1, 2, 5}. In each run, we randomly generate a true label ratio and 20 init points. We run the Adam optimizer for 10000 iterations and compute the TV distance between the estimated ratio and gold ratio. We then compute the mean and variance of TV for each run. The results are as follows.
>
> Mean:
> |Temp|LR=0.1|LR=0.01|LR=0.001|LR=0.0001|
> |-|-|-|-|-|
> |0.1|6.85e-03|9.57e-04|7.21e-04|3.92e-03|
> |0.5|8.30e-04|2.09e-04|2.76e-04|2.74e-02|
> |1|6.82e-04|3.15e-04|2.41e-03|9.83e-02|
> |2|1.56e-04|5.00e-06|2.62e-03|2.35e-01|
> |5|9.80e-05|3.58e-04|2.20e-02|2.50e-01|
>
> Var:
> |Temp|LR=0.1|LR=0.01|LR=0.001|LR=0.0001|
> |-|-|-|-|-|
> |0.1|1.42e-04|3.69e-06|6.00e-06|1.07e-04|
> |0.5|3.03e-06|1.36e-07|1.00e-06|3.61e-04|
> |1|3.00e-06|7.03e-07|3.90e-05|2.06e-03|
> |2|1.27e-07|5.37e-11|3.90e-05|5.83e-03|
> |5|2.83e-08|1.97e-06|2.71e-04|3.31e-03|
>
> We see that when temp <=2 and LR \in {0.1, 0.01, 0.001} (which are typical choices), Alg.1 can consistently achieve < 0.01 TV, suggesting robustness of Alg.1.
> Also, we did not observe any oscillations or collapse during optimization.
>
> > **Comment 3**: CAROT assumes known symbolic partitions. Real-world NESY systems often operate with evolving or noisy taxonomies...
>
> On “known symbolic partitions”: the most well-known NeSY frameworks that study our setting, such as DeepProbLog [32], Scallop [18], NeuroLog [65], assume both the semantics of the neural classifiers outputs, and the symbolic component σ are fixed and known. However, we agree that considering noisy or unknown σ and simultaneously learn the classifier and σ is an important future research direction.
>
> > **Comment 4**: The assumption that miscls depends only on class is strong. Symbolic tasks typically involve multi-factor dependencies (depth of composition, operator type)
>
> First, as we discuss in the conclusion, the assumption that misclassification are dependent only on class is typically adopted in relevant areas, such as noisy learning [68, 41] and weakly-supervised learning [A, B]. Also, a similar assumption that misclassification depends only on class is made by previous theoretical study of our NeSY setting where stylistic factors are independent of the classes, see Section 3 in [36].
> It is important to study such a setup, as it serves as a solid foundation to generalize our techniques to more complex and realistic scenarios.
>
> Second, our bounds do consider “operator type” if that means σ. Could you clarify what it means by the “depth of composition” or point us to theoretical research that characterizes imbalances by taking into account the “depth of composition”?
>
> [A] Learning from partial labels. JMLR, 2011
>
> [B] Wang et al. On learning latent models with multi-instance weak supervision. In NeurIPS, 2023b
>
> > **Comment 5**: The paper assume i.i.d. symbolic structures, yet most NESY include structure dependencies (e.g., scene graphs)
>
> Notice that the i.i.d. assumption is not related to logic programs, which correspond to σ in our formulation and the formulation of NeSy.
> Also, one can still consider the scene graphs or other structures as a whole to be i.i.d. distributed. E.g., in NLP & CV, we consider i.i.d. documents & images, despite they have complex inner structures. In this way, we can still apply the theory developed here. Nevertheless, taking the structure into account may yield sharper bounds. This can be done using known techniques, e.g. [A], that analyze structured predictions assuming bounded dependency.
>
> [A] Stability and Generalization in Structured Prediction. JMLR, 2016
>
> > **Comment 6**: The tightness and operational utility of the theoretical bounds (Prop 3.1–3.4) unclear. Do they guide para selection or batch size decision
>
> Tightness: The bound in Prop. 3.1 is tight in the sense that it can be achieved by constructing a classifier whose label-specific errors are equal to the maximizer of Eq. 2. The bound in Prop. 3.3 can become arbitrarily tight by plugging in any generalization bound for partial risk.
>
> Algorithmic Implication: our bound is independent of any algorithm, which is typical in learning theory. So, it does not directly provide guidance on the hyperparas choice. The advantage of doing this is that it characterizes the intrinsic difficulty/imbalance of the problem itself, which is not caused by the choice of the algorithms.
>
> > **Comment 7**: Evaluation is done on overly synthetic tasks ... considering more complex symbolic datasets like CLEVRER, ProofWriter, GQA
>
> The MAX, SUM, HWF and other benchmarks have been used by the most influential papers in the community, e.g., [32,18,25,36].
>
> Also, we **additionally conducted experiments using VQAR** [18], a VQA benchmark based on GQA with high imbalances. With ILP, the mean balanced accuracy over 3,000 training samples increased from 46.84% (for baseline SL [65]) to to 53.06%. Further details are in our replies to reviewer MSJR’s Comment 2.
>
> > **Comment 8**: Only 3 seeds used, ... For high-variance metrics like balanced accuracy, ≥10 seeds would improve stat reliability
>
> Our Table 5 includes 144 experimental combinations, Table 2 (Appendix) has 72, and we also report results on two additional benchmarks. Given the large number of combos, it is impractical to consider 10 seeds per run. Nevertheless, to address this comment, we additionally provide results with 10 runs on cases with high variance (e.g., MAX-M for $\rho=50$ with $m_P$ = 3K) here.
> Results of balanced accuracy over 10 runs are:
> | |M = 3|M = 4 |M = 5|
> |-|-|-|-|
> |SL|67.72±3.98|70.33±6.88|56.08±1.99|
> |LP(EMP)|78.23±2.71|72.72± 6.34|56.12±1.22|
> |LP(ALG1)|74.01±6.92|72.33±5.27|59.93±3.99|
>
> In the revised version, we will consider more seeds for cases with large variance.
>
> > **Comment 9**: Baseline selection could be expanded ... Even basic class-reweighting schemes are omitted
>
> We kindly disagree that IRM or Group DRO could be considered as baselines in our case, as they focus on different settings: spurious correlations and overparameterization. Also, they assume access to the gold labels during training. So, it would be difficult to adapt them to our setting where the gold labels are unknown.
>
> We considered RECORDS, because it operates at training time and fits our setting where access to gold label is not possible.
>
> > **Comment 10**: No computational benchmarks provided. Training time overhead, LP solver configuration, and memory usage are critical in assessing CAROT’s deployability
>
> We guess the reviewer refers to LP’s deployability since CAROT does not rely on linear programming.
>
> On solver configuration: we simply ran the solver with the default options without any special configuration. This can be confirmed by looking at our code – the solver in utils_algorithms.py and the scripts for reproducing experiments (folder “scripts”).
>
> On computational benchmark: our reply to Reviewer YWhF’ Comment 2 reports all the relevant runtime and memory usage. It shows the baseline training approach and our LP technique are on par in terms of runtime and memory.
>
> > **Comment 11**: Can the author relax i.i.d. assumption to allow structural dependencies? How might this affect Prop 3.1–3.4?
>
> Please see our reply to Comment 5.
>
> > **Comment 12**: What modifications are needed to deploy CAROT on real NESY benchmarks
>
> CAROT is a testing time mitigating technique that modifies the softmax predictions of a classifier given $x$. Hence, it does not require any modifications to be applied to any benchmark.
>
> > **Comment 13**: Have the author considered adapting to hierarchical or graph-structured label spaces?
>
> The new experiments we conducted on a GQA-based benchmark [18], deals with hierarchical labels, e.g., supervision is of the form “Q(O) :- name(O, superclass)”. Further details are in our replies to reviewer MSJR, Comment 2.

---

> > ### Author Response · Authors · 2025-08-04
> >
> > Since the discussion period is ending soon, we are eager to know if there are other points that the reviewer wants to discuss with us.

---

> > ### Comment · Reviewer_qjJg · 2025-08-09
> >
> > LP Constraint Growth:
> > The clarification that the LP grows linearly, not exponentially, via the Tseytin transformation is useful, especially in addressing the concern of constraint growth. However, scalability remains a challenge. While you note that the complexity of enumerating the pre-image is inherent in NeSy frameworks, I would recommend further elaborating on how your approach compares to existing methods in terms of practical scalability. Additionally, discussing alternative approximation strategies, such as sampling-based surrogates, hierarchical grouping of symbolic types, or streaming approximations, could offer valuable insights into how CAROT could handle larger or more complex datasets effectively. Addressing this would provide a more complete understanding of the framework's deployment potential.
> >
> > Stability of Algorithm 1 with Softmax:
> > Your sensitivity analysis is promising, showing robustness in typical settings. However, an ablation study focusing on softmax reparameterization would clarify its impact on optimization stability, especially in skewed conditions.
> >
> > Known Symbolic Partitions:
> > While known symbolic partitions are common in NeSy frameworks, real-world systems often deal with noisy or evolving taxonomies. Discussing this as a future research direction would broaden the framework's applicability.
> >
> > Misclassification Assumption:
> > The assumption that misclassification depends only on class is typical but restrictive in symbolic tasks involving multi-factor dependencies. Relaxing this assumption or providing justification would enhance the paper’s relevance to complex tasks.
> >
> > i.i.d. Assumption:
> > The i.i.d. assumption is standard but limiting for symbolic tasks with dependencies (e.g., scene graphs). Relaxing this assumption would improve generalizability and strengthen theoretical bounds.
> >
> > Theoretical Bound Tightness and Utility:
> > The bounds are tight, but their operational utility remains unclear. Further explanation of how these bounds could guide practical decisions (e.g., hyperparameter tuning) would enhance the framework’s impact.
> >
> > Evaluation on Synthetic Tasks:
> > While synthetic benchmarks are useful, evaluating on more complex datasets like CLEVRER or GQA would better demonstrate CAROT’s real-world applicability.
> >
> > Insufficient Seeds in Experiments:
> > I appreciate the additional experiments with 10 seeds. For high-variance metrics, more seeds would improve statistical reliability and confidence in the results.
> >
> > Baseline Selection and Fairness Benchmarks:
> > Including fairness-focused baselines like class re-weighting schemes would provide a more comprehensive evaluation and demonstrate CAROT’s robustness in fairness-sensitive settings.
> >
> > Computational Benchmarks:
> > More details on solver configuration and real-world deployment considerations would provide necessary context to assess CAROT’s scalability and deployability.
> >
> > That said, reading your rebuttal, I may be inclined to update my rating to borderline accept as I feel the authors have gone through great lengths to adequately address reviewer concerns.

---

### Author Response · Authors · 2025-08-09
**Thank you**

We want to thank the reviewers for their very constructive feedback. We will make sure all their comments are reflected in the new version of our work.

---

### Note · Authors · 2025-08-13

We thank all the reviewers. We summarize the main points that were raised, our responses, and the changes we will incorporate in the CR. **We will make sure our CR reflects all the feedback we got.**

**About the complexity and runtime of our LP technique.**
The reviewers asked to (1) discuss approximation strategies and comment on the complexity of our LP and (2) provide measurements comparing the runtime and memory consumption of our LP against those of the SOTA for varying batch sizes.

About (1):
- Our LP has a linear growth in $\sigma^{-1}$
- Our experiments consider the top-$k$ pre-images to avoid a search space explosion
- The complexity of the SOTA baseline [65] is \#P-complete. Ours is NP-hard
- We discussed several new approximation strategies for solving LPs

About (2), we provided the requested measurements. The results showed that
- The SOTA baseline and our LP are on par in terms of runtime and memory consumption
- The accuracy of our LP formulation increases as the batch size increases, outperforming the SOTA even for batch size 64

We will include the above measurements and the relevant discussion in the CR.

**About benchmarks.** We consider the same benchmarks as the most influential works in the field [32,18,25,36]. Additionally, we ran experiments on VQAR [18], a VQA benchmark based on GQA with hierarchical labels (Reviewer qjJg asked how we perform on hierarchical labels and benchmarks such as GQA). Our experiments showed that we improved the accuracy from 46.84\% to 53.06\% over SOTA. We will include these experiments in the CR.

**About missing re-weighing baselines.** The two suggested baselines, Arjovsky et al., 2019 and Sagawa et al., 2020, are not applicable in our setting: they require access to the gold labels; we assume hidden gold labels. We will include a relevant discussion in the CR.

**About CAROT’s sensitivity.** About sensitivity with regard to softmax reparameterization, we conducted additional experiments with varying learning rates and temperatures. The results showed CAROT’s robustness. We will include these experiments and extend them for varying imbalances in the CR. Regarding the dependence of CAROT on the prior, see Figure 4.

**About reasoning shortcuts (RSs) and label acquisition.** We aim to mitigate imbalances without making extra assumptions as in the RSs work. In the CR, we will elaborate on the relationship with RSs and integration with active learning for label acquisition (i.e., BEARs).

---

### Decision · Program_Chairs · 2025-09-17

**Decision:**

Accept (poster)

**Comment:**

This paper provides a new algorithmic framework for characterizing and mitigating class-wise imbalances in neurosymbolic learning through a LP-based approach. All the reviewers were ultimately enthusiastic about this contribution and in favor of acceptance. The authors are strongly recommended to address all reviewer feedback in the camera-ready version, especially (but not limited to): a) clarifications around the computational complexity of the procedure in theory (e.g. rate of growth of computational complexity in various parameters such as $\sigma$, characterization of the worst-case computational landscape with respect to other state-of-the-art methods), b) add the experiments that they ran in the rebuttal phase around sensitivity to hyperparameters, wallclock runtime compared to competing methods, and memory consumption.